# Quinoa Phenotyping Methodologies: An International Consensus

**DOI:** 10.3390/plants10091759

**Published:** 2021-08-24

**Authors:** Clara S. Stanschewski, Elodie Rey, Gabriele Fiene, Evan B. Craine, Gordon Wellman, Vanessa J. Melino, Dilan S. R. Patiranage, Kasper Johansen, Sandra M. Schmöckel, Daniel Bertero, Helena Oakey, Carla Colque-Little, Irfan Afzal, Sebastian Raubach, Nathan Miller, Jared Streich, Daniel Buchvaldt Amby, Nazgol Emrani, Mark Warmington, Magdi A. A. Mousa, David Wu, Daniel Jacobson, Christian Andreasen, Christian Jung, Kevin Murphy, Didier Bazile, Mark Tester

**Affiliations:** 1Center for Desert Agriculture, Biological and Environmental Sciences and Engineering Division, King Abdullah University of Science and Technology (KAUST), Thuwal 23955, Saudi Arabia; clara.stanschewski@kaust.edu.sa (C.S.S.); elodie.rey@kaust.edu.sa (E.R.); gabriele.fiene@kaust.edu.sa (G.F.); gordon.wellman@kaust.edu.sa (G.W.); vanessa.melino@kaust.edu.sa (V.J.M.); d.sarange@plantbreeding.uni-kiel.de (D.S.R.P.); 2Department of Crop and Soil Sciences, Washington State University, Pullman, WA 99164, USA; evan.craine@wsu.edu (E.B.C.); kmurphy2@wsu.edu (K.M.); 3Plant Breeding Institute, Christian-Albrechts-University of Kiel, 24118 Kiel, Germany; n.emrani@plantbreeding.uni-kiel.de (N.E.); c.jung@plantbreeding.uni-kiel.de (C.J.); 4Water Desalination and Reuse Center, King Abdullah University of Science and Technology (KAUST), Thuwal 23955, Saudi Arabia; kasper.johansen@kaust.edu.sa; 5Department Physiology of Yield Stability, Institute of Crop Science, University of Hohenheim, 70599 Stuttgart, Germany; sandra.schmoeckel@uni-hohenheim.de; 6Department of Plant Production, School of Agriculture, University of Buenos Aires, Buenos Aires C1417DSE, Argentina; bertero@agro.uba.ar; 7Robinson Research Institute, Adelaide Medical School, University of Adelaide, Adelaide, SA 5005, Australia; helena.oakey@adelaide.edu.au; 8Department of Plant and Environmental Sciences, University of Copenhagen, DK-2630 Taastrup, Denmark; cxl@plen.ku.dk (C.C.-L.); amby@plen.ku.dk (D.B.A.); can@plen.ku.dk (C.A.); 9Department of Agronomy, University of Agriculture, Faisalabad 38000, Pakistan; irfanuaf@gmail.com; 10Department of Information and Computational Sciences, The James Hutton Institute, Invergowrie, Dundee AB15 8QH, UK; Sebastian.Raubach@hutton.ac.uk; 11Department of Botany, University of Wisconsin, 430 Lincoln Dr, Madison, WI 53706, USA; ndmill@gmail.com; 12Biosciences, Oak Ridge National Laboratory, Oak Ridge, TN 37831, USA; streichjc@ornl.gov (J.S.); jacobsonda@ornl.gov (D.J.); 13Department of Primary Industries and Regional Development, Agriculture and Food, Kununurra, WA 6743, Australia; Mark.Warmington@dpird.wa.gov.au; 14Department of Arid Land Agriculture, Faculty of Meteorology, Environment and Arid Land Agriculture, King Abdulaziz University, Jeddah 21589, Saudi Arabia; mamousa@kau.edu.sa; 15Department of Vegetables, Faculty of Agriculture, Assiut University, Assiut 71526, Egypt; 16Shanxi Jiaqi Agri-Tech Co., Ltd., Taiyuan 030006, China; quinoaking@vip.163.com; 17CIRAD, UMR SENS, 34398 Montpellier, France; didier.bazile@cirad.fr; 18SENS, CIRAD, IRD, University Paul Valery Montpellier 3, 34090 Montpellier, France

**Keywords:** *Chenopodium quinoa*, descriptors, genetic diversity, scoring card, architecture, panicle, disease, high throughput seed phenotyping, remote sensing, database

## Abstract

Quinoa is a crop originating in the Andes but grown more widely and with the genetic potential for significant further expansion. Due to the phenotypic plasticity of quinoa, varieties need to be assessed across years and multiple locations. To improve comparability among field trials across the globe and to facilitate collaborations, components of the trials need to be kept consistent, including the type and methods of data collected. Here, an internationally open-access framework for phenotyping a wide range of quinoa features is proposed to facilitate the systematic agronomic, physiological and genetic characterization of quinoa for crop adaptation and improvement. Mature plant phenotyping is a central aspect of this paper, including detailed descriptions and the provision of phenotyping cards to facilitate consistency in data collection. High-throughput methods for multi-temporal phenotyping based on remote sensing technologies are described. Tools for higher-throughput post-harvest phenotyping of seeds are presented. A guideline for approaching quinoa field trials including the collection of environmental data and designing layouts with statistical robustness is suggested. To move towards developing resources for quinoa in line with major cereal crops, a database was created. The Quinoa Germinate Platform will serve as a central repository of data for quinoa researchers globally.

## 1. Introduction

Food systems are experiencing intense pressure owing to, among other factors, increasing population and environmental change (such as increases in the frequency and severity of extreme weather events). Changes in environmental conditions are causing changes in where particular crops can be cultivated as well as the types of crops that can be planted in affected areas [1,2]. Uncertainty in climate and weather predictions highlight the need for crops and varieties that are stable across time and regions. Diversification of the crops grown is also important as increasing yield losses are projected to pose a serious threat to food security [3]. These challenges will become ever more paramount with a growing global population. Apart from the concerns of undernourishment in our population, many are not receiving adequate amounts or diversity of micronutrients in their diet; malnutrition is referred to as “hidden hunger” [4]. Dietary intake has been identified as a key factor in the treatment and prevention of numerous non-communicable diseases [5]. The threat posed by climate change to food security, in addition to the concerns of human nutrition, highlights the urgent need for diversification of the food system [6]. Numerous strategies may be used to respond to both hunger and hidden hunger. One such strategy is the reintroduction of genetic diversity to fully domesticated crops to increase variability in the genes responsible for the environmental adaptability, plasticity, and resilience that their wild ancestors still possess. An alternative approach lies in the domestication of plants not yet used in agriculture at the global level and focus breeding efforts in Neglected and Underutilized Species and plants that are only partially domesticated (orphan crops) and those that still harbor considerable genetic variability, which may contribute to improved yield and adaptability. An example of the latter approach is the use of quinoa (*Chenopodium quinoa* Willd.), a crop providing high nutritional content that has become increasingly popular in recent years.

Quinoa has not been domesticated to the same extent as the other major grain crops for global food and agriculture such as wheat, maize, or rice. Because of its cultivation in various agro-ecological conditions over the last few millennia, including mesothermal zones, highlands, salt flats, and subtropical zones of the Andes, quinoa experiences a wide range of challenging environmental conditions, resulting in high genetic diversity and tolerance to a range of biotic and abiotic stresses of this crop [7,8]. Furthermore, the seed is gaining popularity for its high nutritional value, an important feature of the UN’s Sustainable Development Goals [9]. Despite the ability of quinoa to maintain yield under a wide range of environments and the growing worldwide interest in the crop, the primary locations for cultivation are still the countries that have a tradition of growing quinoa: Bolivia, Peru, Colombia, Ecuador, Argentina, and Chile [10]. In recent years, quinoa has been more widely cultivated at sites such as Europe, North America, and China [11]. A primary strategy developed to meet the increasing food demand is the expansion of cultivation through the identification of adaptable and high yielding quinoa cultivars for different agro-ecological zones [12]. Seed nutritional contents and interannual stability of cultivars are traits of much importance in quinoa [13]. Local production will reduce food miles, thereby reducing transportation costs and potentially environmental impacts of quinoa consumption. To achieve this aim, insights into the broad performance spectrum of different varieties grown in a range of environments as well as an improved understanding of the genes underlying traits of agronomic importance are needed.

Recently, genomic tools have emerged to support the development of quinoa germplasm for novel environments. Several novel genomic resources have been developed in the past decade, including bacterial artificial chromosome (BAC), expressed sequence tag (EST) libraries, and DNA-based molecular markers (see [14]), as well as, more recently, a chromosome-scale reference genome sequence of a coastal (or “lowland”) quinoa accession (BioSample Accession Code QQ74) [15] and the resequencing of several wild and cultivated quinoas. Together, these are helping develop genomics-informed breeding programs and genetic studies to accelerate crop improvement [14]. Phenomics (i.e., high-throughput phenotyping) are currently a limiting factor in genetic analyses and genomic prediction, after recent advances in throughput and reduced genotyping costs realized over the past two decades [16]. Quinoa phenotyping strategies, however, have not been standardized. This is one of the limiting factors in the common characterization of quinoa and assessment of its adaptation to different environments, and which limits advancements in quinoa genetic research.

Yield is an important but complex trait in plant breeding because it embodies the link between the cumulative effects of all plant traits and economic value of the crop [17]. Owing to the additive and genetically complex nature of yield, the genetic architecture of yield cannot be easily analyzed, and is usually best analyzed through secondary traits that contribute to yield, such as harvest index and other developmental traits, and photosynthetic parameters and other physiological traits (see [18]). Crop improvement efforts through indirect selection that focuses on secondary traits which are correlated with yield but are potentially monogenic have long been proposed for crops [19,20] and have been successfully used for sunflower [21,22], and many other examples [23,24,25,26,27,28,29]. The assessment of differences in the response of genotypes to different environmental conditions facilitates the development of genotypes that lead to improved phenotypes in a specific environment [30] and also allow insights into the genetic architecture of a trait. To gain insights into complex traits, a study of large numbers of genotyped accessions across multiple environments is required to identify genotype-by-environment (G × E) interactions [31]. This is particularly important in quinoa, due to the large G × E interactions that have been reported [32,33]. For the generation of varieties for different situations, G × E interactions also need to be considered in association with agricultural practices that are shaped by growers and societies. Building collaborations between breeders and growers is important for the adoption of new varieties [34,35].

It is often difficult to determine the pattern of quinoa genotypic responses across environments. The biplot analysis may also provide a powerful solution to alleviate this problem. For effective cultivar evaluation, both the effect of genotype (G) and the interaction G × E must be considered simultaneously [36,37]. A G plus G × E (GGE) biplot was shown to effectively identify the G × E interaction pattern of the data and to show clearly which genotype won in which environments. In addition, the GGE biplot technique is useful in selecting superior genotypes and test environments for a given mega-environment; that is, a group of locations that consistently share the same best genotype or genotypes.

Given the need for data from multiple environments and the high costs incurred from large field trials, the benefits of robust collaborations are clear. In addition to the need for international collaboration for the exchange of genetic material [13], the establishment of an internationally accessible framework for quinoa phenotyping is crucial because it will facilitate the comparison and sharing of data from trials among researchers globally. The previously published “Descriptors for Quinoa and wild relatives” [38] has been a useful guide for the establishment of several phenotypes. We also note the EU Community Plant Variety Office publication protocol for tests on distinctness, uniformity, and stability: *Chenopodium quinoa* [39]. However, there is a need for more detailed explanations of many of the traits, as the current guide leaves scope for interpretation reducing comparability among trials. A lack of good guidelines for the recording of traits or a margin for interpretation, results in uncertainty around data sets that could be reduced through standardization and clear definitions.

Based on our experiences over the past six years since the publication of these descriptors with the International Year of Quinoa, we have identified several useful phenotypes covering the variability expressed by over 1000 quinoa genotypes across multiple environments (including some in which quinoa has not been cultivated extensively) and have adapted the description of some traits to clarify the definitions of the traits. We present this information in this article, resulting from discussion and general agreements among the Quinoa Phenotyping Consortium. Furthermore, this article provides guidelines for the entire process of a field trial, starting with the experimental design, providing advice on crop management decisions, and detailing the minimum environmental data which must be collected. Without standardized metadata information about a trial, experiments cannot be replicated, even when phenotyping standards are followed [40]. 

A consensus on phenotyping methods is presented in this article, starting with phenotyping methods that are performed throughout the growing seasons, including the international standard on the recording of phenological growth stages [41] and options for high-throughput phenotyping using remote sensing. This section is followed by detailed descriptions of the traits that can be assessed in mature plants. These traits are also summarized in phenotyping cards (Appendix A) to aid phenotyping in the field. Next, we focus on the process of harvesting and describe post-harvest phenotyping options and methods. The next step in facilitating the creation of a collaborative network of trials is a platform that allows the viewing of trials undertaken by different researchers globally, through which data can then be easily shared and analyzed. This structure is provided by the quinoa Germinate platform. Thus, this article presents the collective efforts of a large number of quinoa researchers, reflecting the experience gained over many years of working with this crop, and aimed to establish a baseline to approach field trials and move toward an era of accelerated discoveries in the global quinoa research community. The traits presented herein represent the phenotyping scales that are currently in use for exploring the natural diversity across different environments. As quinoa research is progressing, these scales will need to be adapted to match new situations and applications. For this, we are aiming for the Quinoa Phenotyping Consortium to hold an annual meeting to refine the guidelines and procedures with the aim of both increasing the quality and standardization of phenotyping. 

## 2. Quinoa Database

The establishment of an international consensus for our methods in phenotyping quinoa field trials is the first important step in driving international quinoa research forward through allowing comparisons among field trials and facilitating collaborations. An important component of the phenotyping operation is the recording of all information needed for further exploitation of the data. These records need to be complemented with a platform that allows sharing of data generated with the herein described methodologies and facilitates access to the available datasets. 

Contribution of data to this database is of great value to (a) breeders and other people who want to start working with quinoa, (b) other researchers, and (c) to the team that shares their data. Considering the monetary and time cost of field trials, in addition to the phenotypic plasticity of quinoa, access to information on how quinoa varieties perform in different locations is highly valuable in providing decision-making support to growers and individuals who want to start working with quinoa. Access to data from trials in a similar environment would help them to make informed decisions on what varieties might be performing best in their region. For other researchers, as well as the team that contributes the data, the importance of data shared in a database lies in the opportunities for collaborations that it creates. Many analyses require datasets from multiple locations and years. It is through collaborations that the speed of discoveries is accelerated. If a team has not published their data yet and wants to keep it private, it can still be uploaded to the database under a privacy setting through which the data remains accessible only to the person who uploaded it and optionally a defined list of other individuals (i.e., the research team). In addition to these privacy settings, license terms can optionally be assigned to datasets before access to the data is granted. In this case, data would only be accessed after the license has been accepted.

Here, we present a central repository Quinoa Germinate Database. Available online: http://germinate.quinoadb.org (accessed on 15 August 2021), which was created using the Germinate platform [42,43]. The Germinate platform is used by many international organizations, including CIMMYT and data uploaded to Germinate will also become available through the BreedingAPI.

This database allows storing phenotypic and genotypic data as well as germplasm passport data and environmental information. To upload a dataset, users need to first register to the database and data can only be uploaded through the use of standard data templates (https://github.com/germinateplatform/germinate/tree/master/datatemplates accessed on 15 August 2021) to ensure consistency between datasets. For the quinoa community, some of these data templates have been customized to include the information that was highlighted in this paper, including the phenotypes that have been described. These quinoa datasets can be downloaded from the home page of germinate.quinoadb.org. Traits included in the quinoa-traits-data-template.xlsx can be amended to fit your data, but we encourage that all traits uploaded to the Quinoa Germinate Database be trait variables also described in the Quinoa Ontology database of the Crop Ontology (CO_367) of the Generation Challenge Program (GCP) (http://cropontology.org/ accessed on 15 August 2021). Data uploaded using the templates are checked for consistency and correctness by the program before it is imported into the database. A detailed report highlights issues within the data that require fixing before the dataset can be imported. Once added to Germinate, trial data along with climate information can be queried and visualized to discover and highlight patterns and correlations. Data across different trial sites, years, and treatments can be compared to gain a better insight into the effects on performance of germplasm. Customizable lists of germplasm facilitate the comparison of the performance of different accessions, as well as data export of subsets of interest.

All traits described in this manuscript are being uploaded to the Phenotype and Trait Ontologies of the Crop Ontology Curation Tool (https://www.cropontology.org/ accessed on 15 August 2021) to create a Quinoa Ontology database. Any traits measured in Quinoa trials should be included and described in this ontology database. The use of a standard nomenclature for phenotypic traits is another aim in the context of data reproducibility and reusability [44]. The use of standard ontology and recommended methods allows others to quickly interpret and compare data generated from other teams. Hence, we recommend using the described trait variables for quinoa (CO_367), and if a trait is not included in the already described, to add it to the ontology by filing a request (https://www.cropontology.org/add-ontology accessed on 15 August 2021). The ontology spreadsheet can be downloaded from the website and from Quinoa Germinate Database. Available online: http://germinate.quinoadb.org (accessed on 15 August 2021). 

The step of using standardized templates and standard quinoa ontologies ensures that datasets from different groups have the same structure and include the information required by another team to use these data in their analyses. Uploaded datasets can be visualized and analyzed using different tools that are available through the platform. The platform’s integrated tools for data exploration and analysis facilitate the process from data collection to the next exciting steps for new discoveries about quinoa. 

## 3. Germplasm Selection

The quality of an experiment and its usability in an international framework requires careful consideration and planning, with the first step being the selection of genotypes. Based on the proposal of the Plurinational State of Bolivia, quinoa has been recognized by the United Nations (during its General Assembly in New York, 22/12/2011 [45]) as a potentially valuable crop for future generations. In recognition of the considerable genetic diversity created and maintained by the Andean civilization, the International Year of the Quinoa that was declared, with the FAO in charge as the office of secretary [46]. The exceptional genetic diversity resulted from the cultivation and domestication of quinoa for several thousand years in the harsh and diverse environments of the Andes, combined with the tradition of seed exchange by Andean growers. In 2013, Rojas et al. [7,47] estimated that “16,422 accessions of quinoa and its wild relatives, both closer and more distant (*C. quinoa*, *C. album*, *C. berlandieri*, *C. hircinum*, *C. petiolare*, *C. murale*, and *Chenopodium* sp.) were conserved in 59 genebanks distributed in 30 countries”. However, most (88%) of these resources reside in genebanks from the Andean region, for which access is limited to people from those countries [48,49]. Increased international research partnerships with Andean countries is crucial to facilitate access to genetic resources in order to continue quinoa plant breeding and the generation of new varieties which are adapted to new regions. Ex situ collections of 987 and 229 *C. quinoa* accessions are publicly available from the Genebank Information System of the IPK Gatersleben (Available online: https://gbis.ipk-gatersleben.de/ (accessed on 15 August 2021)) and the USDA U.S. National Plant Germplasm System (Available online: https://npgsweb.ars-grin.gov/gringlobal/search.aspx accessed on 15 August 2021)) genebanks, respectively. They include germplasm from 15 countries, mostly the South American countries of Peru, Bolivia, Chile, Ecuador, and Argentina. The extent of duplication within these accessions is uncertain, although likely significant.

To select a panel of accessions representing quinoas genetic diversity, one can, for instance, choose accessions based on their origin, i.e., country/region of origin. Such selection is possible because country/region information is usually documented for accessions available in public genebanks—although it should be noted that the original country of origin can be ambiguous. We found that a substantial proportion of the ex situ collections have more precise information missing about the collection site (region, province, or closest city), preventing them from being selected on the basis of a strict geographical distribution. 

Another option for selecting a subset of quinoa genetic diversity is to consider stable morphological characteristics, such as color and shape of leaf, stem, and seed, as a phenotypic passport for each accession. Such data are not always provided by the germplasm provider, and stability is rarely known, so screening of a large, diverse population is usually required before further selection is possible. 

A third option is to use the phylogenetic relatedness and population structure information provided by recent genetic diversity analyses performed using either DNA molecular marker screening of the different quinoa populations or, more recently, whole-genome resequencing [50,51,52,53,54,55,56,57,58,59]. Whole genome resequencing will be stored in a separate database administered by David Jarvis, BYU (quinoadb.org). Although no comprehensive screening has been performed on the entire collection of quinoa resources available in public gene banks to date, several studies have provided estimations of the genetic diversity of each category of germplasm (in particular, highland vs. coastal (originally designated “sea level”: [60]), the relatedness among accessions, and an estimation of the heterozygosity level at given sites within each population. Cumulatively, these parameters are important for establishing a suitable panel of quinoa genotypes for genetic analysis. 

There is large genetic variation available in quinoa germplasm with a large reservoir of genes for conferring resistance to biotic and abiotic stresses necessary for quinoa adaptation to challenging and changing environments. Developing new plant breeding methods that can maintain a high level of genetic diversity from the population varieties (usually traditional varieties which are often highly heterogeneous material [61]) may confer more stability of seed yields and a higher resilience to the cropping system [62]. The genetic variability in quinoa germplasm is, however, also driving major challenges in fundamental genetic studies. The traditional practice of admixing, together with the possibility of quinoa to outcross, has resulted in extensive allelic richness translating to a relatively high level of genetic heterozygosity in some populations [50,51,52,53,54,55,56,57,58,59]. Owing to this genetic heterozygosity, quinoa genotypes collected in Andean countries and maintained in public gene banks are likely to be highly heterogeneous, necessitating several generations of selfing and single-seed descent to reduce the heterozygosity at potential loci of interest before these materials can be used in genetic studies. Considering the high level of heterozygosity, we recommend a minimum of two generations, and preferably five generations, of self-pollination before initiating any genetic studies to minimize phenotypic heterogeneity and genetic heterozygosity that would interfere with association analyses in genetic studies [63,64]. The use of doubled haploids to obtain homozygosity might also be possible, although this technique has not yet been developed in quinoa. Any accession studied must subsequently be maintained through single-seed descent and given a unique identifier, for it will likely become a selected genotype, genetically distinct from the original material. After several generations and confirmation of homozygosity, seed could then be bulked to facilitate larger trials, in larger plots and/or at multiple locations. 

Another consequence of heterogeneity of seeds in public gene banks is the strong segregation and therefore divergence between plants that are likely to be selected for self-pollination for each quinoa accession in different laboratories worldwide. Ideally, this diversity could be counteracted by growing a subset of seeds from each accession and selecting plants for propagation that represent the majority of the accession. In reality, however, the phenotypic plasticity of quinoa resulting from strong genotype (G) × environment (E) interactions causes quinoa accessions with the same genetic background to produce drastically different phenotypes in different environments [32], making the creation of a phenotypic passport for each genotype difficult. Therefore, caution must be maintained when comparing results generated for a single accession from seed stocks maintained by different genebanks worldwide. These accessions should be considered as genotypes and must be clearly identified with an independent identifier because they might evolve over time into a new commercial variety considering the Distinction-Uniformity-Stability descriptors of the UPOV system (as defined by being material that is new, distinct, and uniform). 

To maintain the relationship among quinoa accessions in different seed stocks worldwide, we propose that the identification of quinoa germplasm is standardized through the use of Digital Object Identifiers (DOIs), allowing each quinoa accession worldwide to be uniquely identified by an alphanumeric string that is assigned by a registration agency such as the Global Information System. Available online: https://ssl.fao.org/glis/ (accessed on 15 August 2021, which provides a persistent link to the location of information about the object on the Internet. This unique identifier co-exists with other identifiers such as those given by the gene banks and allows for unambiguous and permanent identifications of plant genetic resources, which can be exchanged across organizations, and therefore, facilitate the comparison of results obtained by different teams.

## 4. Experimental Design and Crop Management

The statistical design of a plant trial experiment will depend on the aim of the study and will dictate the way in which the data from the experiment are analyzed. Plant breeding and genetics rely thoroughly on large-scale variety trials to generate reliable phenotype data and the scale of those grows continuously [65]. For field trials, spatial designs (i.e., planting layouts in the field where genotypes are replicated and their positions along rows and columns are randomized) are recommended because they allow for better estimation of environmental variation in the field and increase the reliability of the experiments. In general, researchers/breeders should aim for balanced experiments to evaluate the germplasm/varieties in multiple locations and years. Statistical procedures based on mixed models are used to compute fixed effects (best linear unbiased estimate) and random effects (best linear unbiased predictions) from the phenotype data generated by field experiments. If the environment-effect on genotype (G × E) is not significant, no or little difference between BLUE and BLUP was observed. However, when G × E is significant, BLUP was superior to BLUE [66,67]. When a treatment is tested, the number of subjects required is double because treatments always need to be tested alongside control conditions, and both should be fully replicated and separately randomized [68]. There are different options to account for technical constraints that might need to be resolved. For example, a split-plot design might need to be used if a treatment factor, e.g., irrigation treatment, can only be applied to a larger area. A good resource for advice on several different designs is the book *Statistical Methods in Biology: Design and Analysis of Experiments and Regression* [68]. The inclusion of genetic relatedness in the design of early generation trials can also be considered for improvements of selection decisions at early stages in breeding programs [69]. Breeders can also optimize breeding programs to improve efficiency and effectiveness with the help of simulation software such as QU-GENE [70], AlphaSim [71], and DeltaGen [72]. Moreover, analysis can be improved into a new stage by incorporation of genotypic (SNP) and phenotypic data to perform genomic selection (GBLUP) and GWAS [65]. 

In early generation variety trials with large numbers of genotypes and small amounts of seed, it is often not technically feasible to implement a fully replicated design. Partially replicated designs [73], where at least 25% of the genotypes have two replicates, are recommended under such circumstances. If resources permit, fully replicated designs are preferred because the statistical power and confidence in the results increase as the number of replicates of each genotype increase. If breeding values (the estimation of the value that the genes of a variety would have if used as a parent in crosses) [74] are required, a pedigree analysis or a genomic selection approach is needed. Both require generation of a relationship matrix based on either the pedigree or genomic information. Replication in these approaches ensures that the total genetic variation can be partitioned into additive or non-additive variation, which is important for determining breeding values versus commercial values of genotypes [75]; thus, fully replicated trials are preferred. For genome-wide association studies (GWAS) aiming at identifying the genetic variation associated with a particular phenotype, it is important to ensure adequate genetic diversity, allowing enough statistical power to perform an association analysis. Because there is a clear relationship between the effective sample size and the statistical power of an association study, it has been recommended to use a minimum of 100 genotypes to perform GWAS analyses [76], although the minimum sample size also depends on the genetic diversity within the population, the number of markers used in the study, the trait considered, and where the minor QTLs are being sought. Once these data are obtained, replication should be prioritized over the inclusion of a larger number of accessions. In limited space, the inclusion of more genotypes but fewer replicates comes at the expense of obtaining an accurate estimation of their phenotypic performance, which in turn reduces the power of SNP-loci association calculations, and might lead to false positives. A good balance can be achieved with partially replicated designs which combine the advantages of high replication levels for a subset of the genotypes and a large enough genetic diversity in the panel of genotypes. 

DiGGer [77] is a useful package for R software that can be used to create partially replicated and fully replicated spatial designs. It can be downloaded from NSW DPI Biometrics software download page. Available online: http://nswdpibiom.org/austatgen/software/ (accessed on 15 August 2021). Once in the R version 4.0 software environment, the file can be installed from the downloaded zip file (e.g., “digger.zip”) using the following commands:

install.packages(“C:/path/DiGGer.zip”, repos = NULL, type = “source”)

where “C:/path/” indicates the path to the folder in which the zip file has been saved. The downloaded zip file contains a manual with examples of the code required to create these designs. An example of a fully replicated and a partially replicated trial created in DiGGer is shown in Figure 1A,B respectively. Other packages with many useful functions are agricolae: Statistical Procedures for Agricultural Research (Available online: https://cran.r-project.org/web/packages/agricolae/index.html (accessed on 15 August 2021)) and ASReml-R. Available online: https://www.vsni.co.uk/software/asreml-r (accessed on 15 August 2021).

Although not included in the trial design examples shown, it is very important to include the rows and columns of border plants that are needed around the field to provide a “buffer area” for the experiment, thereby reducing the environmental border effects. This buffer area should be at least 1 m wide and should be planted with one or two well-performing genotypes. Border effects are also observed at the plot level, making the high phenotypic plasticity of quinoa immediately apparent. Plants on the edges of the plot commonly show higher levels of branching, which is probably a response to the increased space and resource availability. These plants can also differ greatly in height from plants in the middle of the plot, a phenomenon which may depend on their exposure to nitrogen and other nutrients, water, and varying ratios of red to far-red light. If there are maturity or height variations among genotypes, neighboring plots may also cause shading, which is an issue to be considered when planning the plot size.

Therefore, it is recommended to have plot sizes of approximately 4 m^2^ to allow a “plot in plot” design. When recording observations, researchers can disregard the plants on the outer edges of the plot, which occupy approximately 0.5 m on each side, and focus only on the plants in the middle of each plot. The phenotypes observed in the middle of a plot are the phenotypes of interest because the proportion of plants along the edge becomes negligible when plants are grown in large fields at a commercial scale. Border rows may be used for bagging and seed multiplication. However, if bagging is not adequately performed, heterogeneous seeds may be obtained, which should not be used in future trials.

Regarding yield predictions, larger plots are more beneficial, and bed sizes such as 5 m long × 3 m wide (12 rows, 25 cm between rows) = 15 m^2^ are recommended to ensure a more accurate representation of the behavior and expected yield from an accession if it were cultivated in a large field. A 10 cm spacing between plants within a row equates to 600 plants per plot. If 500 plots are used, the field area required for the plots is 7500 m^2^. The total area required will be larger, after adding in the area needed for spacing between plots, and the inclusion of border plots around the field. Smaller plots are often required owing to individual constraints. We found plots sized 2 m × 2 m to be a practical size for multi-environment trials quantifying phenotypic traits to assess a large set of several hundred accessions, but plots this small should not be used for accurate measures of yield.

In any trial, growing conditions should be as close as possible to the actual condition under which the plants will be grown commercially, particularly if investigating yield or traits closely related to yield [78]. These conditions involve minimization of both abiotic stresses (such as water and nutrients) and biotic stresses (such as pests, weeds, and diseases). If stress effects are important, these conditions must be implemented in plants grown under otherwise optimal conditions to allow quantification of the specific responses to stress. Optimal conditions and crop management approaches are largely dependent on location, soil types, and genotypes. The following section provides some suggestions for planting and growing conditions. More detailed recommendations for specific situations can be found in the book “Quinoa: Improvement and Sustainable Production” [79]. Although there are no clear guidelines for some of the following crop management aspects yet, a research gap that needs to be addressed, it is advisable to follow strategies that are successful for others growing quinoa under similar environmental conditions and soil types [13,80].

### 4.1. Planting

Planting density has a significant effect on the phenotypes of quinoa (see, e.g., [81]), specifically on their branching habit, and should therefore be optimized to suit the cropping system. Soil types and irrigation systems are important factors to consider; for example, factors such as furrow irrigation can restrict row width. When assessing the effect of plant density, it is important to consider two distances: the distance between rows of a plot, plus the distance between plants in a row. Studies that have assessed the effects of planting density on quinoa phenotypes were conducted in a specific environment. Due to the use of different methodologies and genotypes, comparisons between environments are not possible to draw. It appears that the optimal planting density depends on multiple environmental and management factors, such as weed management, as well as genotype [82]. Considering the significant phenotypic plasticity of quinoa in response to planting density, it is clear that trials with similar planting densities are better in multi-environment analyses and that planting density always needs to be recorded. Moreover, plots with poor emergence should be recorded, and might need to be excluded from analyses. When assessing hundreds of genotypes in one trial, only one uniform planting density can be used, unless planting density is a criterion being tested. 

Current and future cropping systems aim for the production of high yielding varieties with highly nutritive seeds, requiring low fertilizer and phytosanitary inputs. Optimum sowing density can vary depending on quinoa variety, which in turn can differ according to plant and panicle architectures, seed size, sowing technique (broadcast, rows, or grooves), and agroecosystem. Quinoa sown at high densities grow into less robust and smaller plants, with lower yield per plant, than those planted at low densities. However, the planting of too few plants per unit area may result in branched plants that may not mature before the first frost, and it may also provide more space for the growth of weeds. Therefore, Aguilar and Jacobsen, 2003 [83], have recommended a density of 40 plants per m^2^, with 10 cm between plants in a row and 25 cm between rows, depending on practical details of the system. Quinoa seed viability is sometimes low; therefore, it is prudent to plant more seed (~3 kg/Ha without taking seed size into consideration), followed by a thinning of plants to achieve the desired distances between plants. Thinning (removing excess seedlings) should be performed when plants are 5 to 10 cm high, after the plants are established, but before signs of intense competition for light, such as elongated seedlings, are detected. Sowing a lot of seed followed by thinning to achieve the desired plant density is a method often popular in a field situation that does not allow the recording of seed viability in the field. Hence, germination rate of the seeds from each seed batched used in the field trial should be tested in petri dishes to calculate seed germination rate (%) and other germination parameters. This also allows more precise calculation of the amount of seed that needs to be sown out (see, e.g., [84]). 

The above recommendations relate to ordinary flat fields. In the Andes highlands, farmers have through generations developed a completely different strategy where they sow numerous seeds in holes that are distant from each other. The outermost plants protect the plants in the middle from flying sands and strong winds while the holes also collect water. 

Planting may be performed by hand, although this approach is labor-intensive and is not feasible for large trials. Alternatively, a hand push seeder can be used. The use of a mechanical seed planter (such as a cone seeder) allows even higher throughput, although this requires a skilled operator to ensure that seeds from one genotype are not carried over into adjacent plots. Seeds are planted at shallow depths. A recommendation is to plant at a depth that is three times the diameter of the seed. With seeds between 1.5 and 2 mm in diameter, the planting depth should be approximately 4.5–6 mm. Recommended planting depths can vary with location and soil types. Soil compaction is important because quinoa seeds germinate better in looser soils, so germination rates are better in sandy soils than in heavy clay soils [85]. Hydromorphic soil types are problematic for growing quinoa due to a high sensitivity to waterlogging [86]. Waterlogging was also found to negatively impact the percentage of emergence when seeds were planted too deeply, whereas shallow-planted seeds may be subject to drying [87]. A uniform planting depth of the seeds is important to reduce the risk of uneven field emergence.

Water availability and other environmental constraints can be considered in the decisions around the season for cultivation. Depending on environmental factors or for consideration of crop rotations, quinoa can be used as a winter or spring crop. The sowing date requires careful consideration as this decision impacts growth and productivity of the crop. Due to a range of factors from soil temperature for seed germination through to high temperatures inhibiting grain fill [88]. Consideration could also be given to staggering planting to account for differing times to maturity depending on the traits being measured in the experiment. 

### 4.2. Irrigation

Irrigation is known to affect several aspects of quinoa phenotypes, from plant height [89] to seed saponin content [90] and yield [91,92]. Irrigation needs for optimal growth depend on soil type and environmental conditions and should therefore be calculated for each location. The irrigation requirements for quinoa can be estimated using the crop coefficients (K_c_) for quinoa, as described below.

To calculate the evapotranspiration rates, a reference evapotranspiration rate (ET_0_) is first calculated from a range of meteorological parameters using the only standard accepted method by the FAO, the Penman–Monteith equation [93]. To facilitate the calculation of ET_0_, a calculator has been developed by the FAO for Windows OS [94]. Next, water requirements can be calculated by multiplying the ET_0_ with quinoa crop coefficients (K_c_ × ET_0_), and finally, irrigation amounts are planned by subtracting any rainfall from the water requirements that were calculated. The requirements differ with growth stages, considering that the principal growth stages 6 (flowering) and 8 (ripening) are most sensitive to drought stress [95]. Rainfed crops may therefore need additional irrigation at several points of the growing cycle: before sowing, at the beginning of flowering, and at the start of seed filling. Proposed K_c_ values also reflect the differing requirements by providing three different values to choose from: one of the following three K_c_ values may be used to make a crude estimate of water requirements: 0.52 for initial stages, 1.00 for mid-season stages, and 0.70 during the principal growth stage 9 senescence [91], although compare [96]). More accurate estimates of water use can be performed by using a growth model relating the dynamics of leaf area and radiation interception. Integrated crop management tools such as the SALTMED model can be used for informing irrigation strategies, and has been calibrated and used for quinoa [6,97,98]. The K_c_ values used in these three model applications differed among them, suggesting that different K_c_ values yield better model predictions in different environments and for different genotypes. For example, planting densities were found to greatly affect K_c_ values, varying to such an extent that a single K_c_ value was difficult to assign [82]. 

Owing to strong phenotypic responses of quinoa to water availability, it is also crucial that the land used for the experiment is laser leveled before sowing to reduce the heterogeneity in the amount of water that plants receive across the field. Types of irrigation vary depending on local environmental conditions and technical possibilities. In any case, it is important to record details of irrigation schedules and the amount of water used. Even though quinoa is considered a facultative halophyte, salinity levels of the irrigation water are important to measure with precision as well, not only when salinity is a treatment. 

### 4.3. Fertilization

There has been little research on the fertilization of quinoa. Most publications provide information on local recommendations but lack thorough physiological and biochemical characterization across genotypes. Several studies focus only on varying inorganic nitrogen supply following the conventional method for determination of optimal nitrogen availability by measuring responses according to yield [99] without examining growth responses to other important macro- and micro-nutrients or organic sources of these nutrients. Quinoa yield has been shown to positively respond to nitrogen supply [96,100,101]; however, it is important, when breeding quinoa for smallholder growers, to consider the selection of genotypes that can maintain their yield under nitrogen limiting conditions. Interestingly, quinoa cv. Titicaca has been shown to maintain size, weight, and nitrogen content of their seeds irrespective of the nitrogen supply [102,103]. 

### 4.4. Weeding, Pest and Disease Controls

There have not yet been any effective herbicides developed for quinoa and the crop is mainly cultivated under organic practices; therefore, weeding often needs to be performed by hand. Weeding is important because weeds compete with quinoa plants for nutrients, light, and water, confounding the results of systematic investigations on the culture of quinoa and potentially reducing yield. If machine weeding is available, spacing between rows/plots needs to be sufficient to enable movement of machinery. 

For scientific studies in-field, preventative management of pests and diseases need to be conducted. For this, it is better to apply treatment on a regular schedule rather than waiting for problems to appear. Of course, for trials testing resistance to pests and diseases, such a regimen should not be conducted, nor for commercial fields using integrated pest management. In several countries, no approved products for quinoa are available; therefore, management must often be based on recommendations for similar crops such as beet, chard, or spinach. For preventing loss in quinoa yields, it is essential to be aware of the diseases and pest that may occur, but in the case of quinoa, there is high uncertainty about local ecological interactions when adapting the crop to new agro-ecological zones and environments. The only manual on quinoa pest and disease [104] was developed for Andean conditions and it was not adapted for different conditions of cultivation, nor translated into English. This point implies that we need entomologists and plant pathologists within the research teams for controlling and preventing damages in trials. Digital tools such as Platix [105] to detect diseases and foliar disorders are becoming increasingly powerful and can supplement expert input. 

## 5. Environmental Variables

Although crop management techniques can be used to influence some environmental variables, such as when irrigation regimes are planned according to rainfall, virtually no environmental variables can be controlled in a field experiment. These variables include soil drainage, soil pH, and microtopographic effects, causing pooling in some locations [106,107]. However, environmental variables greatly affect phenotypes, and the degree of plasticity of a genotype to adapt to environmental conditions varies greatly. Thus, consideration of G × E interactions are of great importance in matching genotypes with the appropriate environments for achieving maximum yield. The investigation of these interactions also facilitates the uncovering of genetic correlations by plant geneticists [108]. To gain clearer insight into the input of genetic factors to a trait, potential variability introduced by environment (E) and management (M) needs to be reduced as much as possible. This can be achieved if as many variables as is reasonably possible are monitored. To limit the effects of management on the variables measured in a trial, the management practices should be harmonized by adopting the recommendations from the section above. Environmental variables can vary substantially among and even within the plots in a field. Therefore, it is important that researchers try to collect at least a minimal set of variables for a field site from a nearby weather station. The most important variables that should be collected for each trial are summarized in Table 1. 

Effective nutrient and soil management relies on data from the testing of soil cores by internationally accredited laboratories. A soil core sample is collected using a hollow steel tube called a “core drill”, which may be up to 40 cm in depth. It is best if cores from 0–20 cm deep are separated from those taken at 20–40 cm depth. There is considerable spatial variability in the physical and chemical properties of soil both horizontally and vertically, and therefore, large sampling regimes are recommended. Multiple core samples (25–30) should be collected at random sites from multiple, well-defined locations such as fence lines, tree lines, hills, or GPS coordinates. 

A more strategic approach involves defining “zones” within the field where variations in management practices are predicted to be necessary owing to differences in slope or soil color or areas in which growth has been shown to vary in previous years. A good visual example is provided in [109]. This strategy can reduce sampling time by using more “cluster” sampling, i.e., using five cores per zone. Soil sampling depth depends on the rooting depth of quinoa accessions used in a specific soil and can be assessed using a soil core drill in the plot and looking for roots within the soil core. The soil type affects numerous factors, including the soil’s water holding capacity, nutrient storage, and aeration. These factors affect crop productivity and phenotype. Soil texture is an important factor to include in the environmental information [110]. The USDA provides soil texture calculators to define a single point texture class based on the percentage of sand, silt, and clay. This calculator is available at the following URL: https://www.nrcs.usda.gov/wps/portal/nrcs/detail/soils/survey/?cid=nrcs142p2_054167 (accessed 15 August 2021). 

Proximity to the study perimeter can also cause G × E effects on plant phenotypes. Outer plants incur the brunt of wind, causing high rates of evapotranspiration. In a randomized trial assessing diversity of phenotypic variation across the population, neighboring plots can impact the conditions of each plot. It is possible that larger and more heavily branched plants require more water than smaller plants, and therefore, irrigation systems could over- and under-water plants of different sizes. This issue again highlights the importance of replication in experimental design and the importance of measurement of environmental parameters. The outputs from measurement devices and plot locations in association models can be used as covariates as a broad control for known or unknown factors [106].

Depending on the aim of a trial, more detailed environmental measurements might be needed. Humidity and temperature, which can be combined into a heat index, can account for within-site environmental variation in studies of disease traits because fungal species generally prefer humid conditions. The cost of temperature–humidity meters ranges from less than USD 100 to several hundred dollars. Using plot position, i.e., two columns of vectors of site grid positions, as covariates in spatial models can account for some environmental effects. However, better corrections are achieved with measurements of environmental information as covariates. Small hand-held temperature–humidity devices such as Extech EA20 (Extech Instruments, Nashua, NH, USA, available online: www.extech.com (accessed on 15 August 2021)) or a UNI-T UT333 (Uni-Trend Technology, Dongguan City, China, available online: www.uni-trend.com (accessed on 15 August 2021)) and others may be used for measurements at a plot level. Trials examining crop responses to water also require measurements for atmospheric vapor pressure deficit (VPD), which is related to evaporative demand and thereby driving plant water transport. The higher the pressure deficit, the higher the water stress experienced by the plant [111,112].

The amount of soil moisture per plot can be used to correct environmental variations when water use efficiency and photosynthetic traits are measured. Soil moisture measurements for individual plots can be used as covariates to correct environmental variation in water use efficiency and photosynthetic studies, as can thermal measurements in heat stress conditions. Soil moisture sensors also have a wide range of costs and can provide much needed assessment of local soil factors such as temperature, pH, and water content that could drive some of the phenotypic variation in a study. Several inexpensive devices are available; however, advanced systems can cost approximately USD 1000, such as the Vegetronix VG-Meter-200 (Vegetronix, Riverton, Utah, USA, available online at: www.vegetronix.com (accessed on 15 August 2021)) is a mid-range soil data logger or APERA PH8500-SL (Apera Instruments, Wuppertal, Germany, available online at: www.aperainst.com (accessed on 15 August 2021)). Gravimetric measurements of soil water content can also give some indications of water availability for plants and can in some conditions be the only method practically feasible. Sensors measuring soil moisture frequently overestimate soil humidity in soils with high levels of iron [113] and in this case, water content can be measured gravimetrically. 

Crop growth simulation models such as those of the Decision Support System for Agrotechnology Transfer (DSSAT) can be used to investigate G × E interactions. Crop models can be used to facilitate crop management decisions by evaluating multiple scenarios. The integration of quinoa into the DSSAT system was initiated with the calibration of the CROPGRO template for quinoa [114]. However, more work on getting DSSAT models calibrated and usable for quinoa is required. Other crop growth models or crop water models such as SALTMED and AquaCrop are good resources for quinoa and already applied (e.g., [97,98,115,116,117,118,119]). The minimal requirements necessary to use these models must be considered (for AquaCrop see [120]; [121]; for SALTMED see [122]). 

To facilitate the reuse of data and establishment of multiple local networks of phenotypic data, Minimum Information About a Plant Phenotyping Experiment (MIAPPE) metadata standard for plant phenotyping must be followed [40]. A MIAPPE spreadsheet template is published online for guidance, MIAPPE available online: https://github.com/MIAPPE/MIAPPE (accessed on 15 August 2021). Alternatively, we recommend downloading the data templates created for the Germinate platform, which are created in accordance with metadata standards, data templates available online: https://github.com/germinateplatform/germinate/tree/master/datatemplates (accessed on 15 August 2021). Templates specifically for quinoa are available at the Quinoa Germinate Database. Available online: http://germinate.quinoadb.org (accessed on 15 August 2021) In this manuscript (see Section 9 for more details), we also offer a platform for sharing quinoa datasets using the Germinate database structure [42,43]. To use this structure, the templates provided should be filled in with the relevant information. Therefore, it would be best to start using such templates from the initial planning of a trial and then continue with phenotypic observations. 

The above list of environmental parameters is comprehensive and will be difficult to achieve for all experimental sites. Pragmatic decisions will often need to be made and can be guided by both the particular properties of the site and the scientific questions being addressed. 

## 6. Observations during Growth

There are several measurements that can be conducted throughout the growing period.

### 6.1. Phenology over Time

Clearly defined phenological stages are of great importance for reproducible phenotyping. Multiple studies have investigated and described phenological stages in quinoa [41,123,124,125,126,127]. These studies have provided valuable information about the characterization of the crop; however, only one study has followed the complete international scale system proposed by the Biologische Bundesanstalt Bundessortenamt und Chemische Industrie (BBCH). Sosa-Zuniga et al., (2017) [41] provided the most recent and complete description according to the BBCH guidelines. Here, the main principal growth stages and their relevance to phenotyping are briefly summarized. An overview of the main growth stages is also shown in the phenotyping cards (Appendix A). 

Sosa-Zuniga et al., (2017) [41] describe eight major growth stages. All developmental stages from seed germination until cotyledon emergence belong to principal growth stage 0 (BBCH 00–09). The sowing date is the most critical part of this stage because it should be optimized for local conditions. The next principal growth stage 1 covers all stages of leaf development (BBCH 10–19). Fully emerged cotyledons can be observed in BBCH 10. This stage is considered the field emergence stage, and it is necessary to record both the time and the emergence percentage. The emergence percentage can be measured by assessing a randomly selected 1 m^2^ area from the middle part of a plot. At BBCH 11, true leaves develop, and the time from stage 10 to 11 is essential for estimating the early seedling vigor of quinoa accessions. 

At principal growth stage 2, branches start to grow from the basal leaves. This stage can occur before the principal growth stage 5, depending on the accession. The inflorescence emergence stage (principal growth stage 5) is one of the most critical stages in quinoa development. At the initial inflorescence emergence stage, floral buds are enclosed in leaves (BBCH 50) and are thus difficult to phenotype. At BBCH 51, buds are visible from the top but still surrounded by the leaf primordia. This stage should be recorded and can be represented as days to inflorescence emergence. 

Principal growth stage 6 in quinoa is its flowering stage. Because yield and many other agronomically important traits are highly correlated with flowering, this stage is vital. According to Sosa-Zuniga et al., (2017) [41], stage 6 is divided into three BBCH stages. BBCH 60 is the beginning of anthesis, which is marked by flowers opening and extruding their anthers. BBCH 67 is the beginning of senescence of anthers. BBCH 69 is the completion of senescence of anthers. The length of time between BBCH 60 and BBCH 69 depends on the accession and can vary widely. Phenotyping of BBCH 67 and 69 can be imprecise; therefore, reproducible phenotyping attempts in the flowering stage should be focused on recording BBCH 60. This stage is used to score the flowering time of an accession. A panicle can be classified as BBCH 60 as soon as at least one flower with extruded anthers is visible. When scoring flowering time in the field, a plot may be scored for the onset of anthesis, once >50% of plants in the plot have reached this stage. The number of days to flowering among quinoa accessions is highly diverse. 

The next principal growth stage is the fruit development stage BBCH 70. At this stage, ovary thickening occurs and can be identified by naked eye, as shown in the phenotyping cards (Appendix A). Depending on the accession, BBCH 70 and late flowering stages could overlap. Therefore, it is challenging to record this stage, and these difficulties might result in imprecise phenotyping. Nonetheless, this is a crucial stage because it is the beginning of the seed set, and therefore, careful and frequent observation should be performed to record the BBCH 70 stage. 

Principal growth stage 8 comprises three ripening stages: BBCH 81, 85, and 89. At BBCH 81, onset of seed filling can be observed, and seeds at this stage appear as milky grains owing to the nature of soft fruits. Thick and fully ripened grains are present in the stages BBCH 85 and 89, respectively. When scoring plants at this principal growth stage, it is important to carefully examine the seeds by sampling some from the upper middle half of the panicle and crushing the seed by hand, typically with a fingernail. Quinoa accessions show varying degrees of ripening panicles, and a single panicle may show all stages at a single time point, indicating a highly unsynchronized panicle ripening. When the complete panicle is at BBCH 89, it is ready to harvest, but the plant is still not fully senesced. Mechanical harvesting may be difficult at this stage. BBCH 89 is considered the stage of physiological maturity. Sometimes, BBCH 89 is not possible to score, particularly with day-length sensitive plants that can exhibit regrowth in the panicle.

Senescence is the next growth stage (9)—although sometimes this can occur before BBCH 89. Five BBCH scores are assigned to this stage. Senescence starts from the basal leaves (BBCH 91) and spreads upward in a plant. At BBCH 95, all leaves are dead, and the stems turn yellow to beige in color. Plants are ready for harvest when the whole plant is dead at BBCH 99. However, depending on the accession, it may be advisable to harvest before plants reach this final stage to prevent seed loss from shattering or feeding birds.

For the precise recording of principal growth stages, observations must be performed two to three times a week. All plots must be assessed on the same day, and observations should be evenly spaced throughout the growth cycle. At least 50% of the plants in the inner part of a plot should have reached the corresponding stage for the recording of the growth stage of the plot, as judged overall by eye. In practice, recording all stages may not be feasible, especially when the trials are large. In this case, it is necessary to record at least the sowing date, field emergence (BBCH 09), emergence of first true leaves (BBCH 11), floral bud emergence (BBCH 51), anthesis (BBCH 60), physiological maturity (BBCH 89), and harvest date.

Note that this approach for the scoring of phenology is distinct to the measuring of more specific (often physiological) traits, where it is usually best for at least 3–5 plant per plot be measured and marked for resampling of the same plants (not randomly selected plants each time). It is best if these plants are within the final harvest sector or at least comply with requisites for sampling area (e.g., not border plants).

Apart from recording the timing at which each genotype reaches the selected phenological growth stages, another phenotype that must be assessed during growth are leaf area-related measurements, such as radiation interception efficiency. Because the last main stem leaves of quinoa plants appear around the beginning of anthesis (BBCH 60), whereas those on branches continue expanding until the end of flowering [127], the best growth stage to count leaf numbers and areas is after the anthesis growth stage (BBCH 70). Of course, an ideal approach would be to count leaf numbers as soon as leaves emerge throughout the growth period because by anthesis, many would have fallen from the plant. However, this is highly labor-intensive and often not feasible. 

### 6.2. Radiation Capture and Efficiency of Use

The yield potential of a crop in field trials is correlated with the amount of photosynthetically active radiation (PAR) available and the plants’ efficiency in capturing it [128]. Variability in radiation capture owing to differences in the architectural traits of genotypes and efficiency of use of radiation in response to environmental conditions have been proposed as heritable traits which could be used to reduce the complexity of phenotypic responses to the environment. The results of genetic analyses of these traits, in combination with environmental and genotype information, can be used in yield simulations. This approach requires data from experiments performed in different years and locations for the validation of the simulation results, although most other traits have to be assessed in the context of much larger networks of experiments across multiple years and locations [129,130]. 

Using a resource capture and efficiency approach, biomass accumulation by a crop can be modeled as follows:$$biomass=resource\;availability\times resource\;capture\;\left(or\;uptake\right)\;efficiency\times resource\;use\;efficiency$$
where the resource can be radiation, water, nitrogen, or another nutrient. From a management perspective, radiation availability can be partially achieved by the selection of sowing dates and location and selection of genotypes with suitable cycle durations, whereas water and nutrients can be managed by altering irrigation or fertilization practices. 

One of the main determinants of the uptake of any resource is radiation interception efficiency, i.e., the proportion of incident radiation intercepted by the crop. This property affects both radiation capture and the partitioning of water use between potential transpiration and evaporation [131]. Interception efficiency (IE) is determined by the leaf area index (LAI), the leaf area per unit soil area, and *k*, the light extinction coefficient. The association between IE and *k* is described by Beer’s law [132] as follows:$$IE=1^{\left(-k\times LAI\right)}$$

The association between LAI and IE is affected by variation in the light extinction coefficient *k.* In quinoa, variation was detected according to genotype and plant density, with a positive correlation between *k* and plant density, suggesting rearrangement of leaves when the density was modified. However, a common *k* of 0.59, resulting in a critical (95% IE) LAI value of 5.09 can be used across genotypes and densities [127]. If this *k* value is assumed, IE can be calculated with LAI estimations from emergence to critical LAI. Estimations of LAI can be obtained through destructive leaf sampling or a portable, non-destructive, Plant Canopy Analyzer. As these two methods are low in throughput or not calibrated for quinoa, the use of remote sensing, as described later in this section, is a better alternative.

Leaf area indices can be calculated from measurements of IE, which can be estimated using a ceptometer, an instrument for measuring PAR (see details about its estimation in [127]), and by taking measurements above and below the canopy, the amount of light that is not used by the plant for photosynthesis can be determined. Results can be used as an estimation of radiation capture, integrating LAI development, senescence, and *k* variation aspects. Measurements using a ceptometer are time-consuming; however, because it can only be used for a couple of hours around midday, and for each replicate, six measurements need to be taken, two above the canopy to determine incident irradiation and four below the canopy along the ground to calculate the amount of intercepted radiation [133]. 

The last component of the biomass accumulation equation, resource use efficiency, which equals radiation use efficiency (RUE, g m^−2^ MJ^−1^ or [g mol m^−2^ d^−1^ PFFD^−1^]; PPFD: photosynthetic photon flux density) from a carbon balance perspective, can also be calculated from IE measurements. Similar to LAI, RUE is associated with accumulated intercepted PAR (∑ incident PAR × IE over the period among samplings). RUE might not be stable during the crop cycle of quinoa. Ruiz and Bertero (2008) [127] detected two different RUE estimates from emergence to end of flowering (BBCH 70), an initial low value of 1.25 g m^−2^ MJ intercepted PPFD^−1^ and a second, higher RUE value of 2.78 g m^−2^ MJ intercepted PPFD^−1^ with a breakpoint at 107 MJ m^−2^ [127]. This experiment was run using a mid-winter sowing time, and no similar differences in RUE across the cycle were detected for spring sowings [134]. Low RUEs under low temperature conditions are expected to occur in several quinoa growing environments, such as the high Andes, during early spring in Northern Europe, or autumn and winter in Mediterranean environments, which partially explains the low initial growth observed for quinoa in many environments. 

However, the approaches for LAI and RUE measurements described here are not feasible for application in large field trials owing to several limitations, which make them low in throughput. This process is time-consuming and labor-intensive and also affects the quality of the data collected because measurements must be taken over a longer time span. Instead, unmanned aerial vehicles (UAVs) deployed with multi-spectral sensors may be used to collect image data for the entire trial area to estimate LAI of all quinoa plots. However, low-throughput field-based methods for measuring LAI are still required for modelling LAI from UAV-based imagery and for independent validation of the image-derived LAI measurements Estimates for RUE or radiation capture can also be obtained from the Photochemical Reflectance Index (PRI), which can be extracted from spectroradiometers (PRI) or normalized difference vegetation index (NDVI) estimates [135]. Several spectral reflectance indices can be obtained at a high throughput through the use of UAVs. Of these indices, NDVI and green normalized difference vegetation index (GNDVI) have been identified as the most informative for quinoa when assessing irrigation treatment effects [136,137]. Furthermore, spectral reflectance measurements have been identified as the most effective tool in the assessment of disease impacts in quinoa [138]. 

### 6.3. Unmanned Aerial Vehicle-Based Phenotyping

Phenotyping methods have been considered a bottleneck in plant genetic studies [16]; therefore, it is important to quickly progress to non-destructive high-throughput and high-precision options for phenotypic data collection. The use of UAV technology is emerging to be one of the most promising solutions. Recent advances in UAV technology and miniaturization of mountable sensors have facilitated accurate, consistent, and expansive high-throughput phenotyping of crops [139] and standardized data collection approaches and image processing workflows [140]. The use of UAV-derived data for crop phenotyping may increase the amount of data collected, frequency of data acquisition, consistency of information extracted, and the ability to undertake retrospective studies, while significantly reducing human labor. Red-Green-Blue (RGB) and multispectral cameras are used most frequently for crop phenotyping studies because of their relatively low costs and ease of use; however, Light Detection and Ranging (LiDAR), thermal infrared, and hyperspectral sensors have also been used to collect information on phenotypic traits [141]. UAV-based RGB and multispectral cameras have been found to be suitable for plant height estimation of maize [142], yield prediction of tomatoes [143], plant vigor assessment of barley [144], detection of pathogens, e.g., cotton root rot disease [145], yellow rust disease in winter wheat [146], and maize streak virus [147], and mapping of growth patterns of winter wheat [148]. UAV-captured data can also be used to calculate a crop water stress index [149] where leaf temperature is normalized using environmental conditions measured around the experiment along with leaf temperature. The use of digital photogrammetric processing approaches such as structure-from-motion and multi-view stereo reconstruction of overlapping optical images allows detailed surface elevation characterization and three-dimensional models to be generated [150]. Although LiDAR sensors have greater plant canopy penetration capabilities and provide height information with higher accuracy than photogrammetrically processed optical data, their weight, costs, and range currently limit their operational use in the agricultural industry [151].

UAV-derived thermal data have become increasingly popular for use in phenotyping applications to monitor canopy temperature, detect plant disease, and estimate yield [152]. However, camera effects such as vignetting, camera warming, and temperature drift as well as meteorological conditions such as ambient temperature, wind, and wind direction often affect the accuracy of UAV-derived at-surface temperature measurements [153,154]. Malbeteau et al., (accepted) [155] provide practical information and examples of how to overcome issues related to dynamic temperatures and wind effects during thermal UAV data collection to improve data consistency and accuracy of UAV-based orthomosaics of a field trial designed for phenotyping of tomato plants. UAV-based hyperspectral imagery contains hundreds of spectral bands that allow the collection of detailed spectral information on phenotypic traits [156]. Examples of agricultural hyperspectral data applications include estimation of chlorophyll, mapping of plant disease, detection of nutrient status, and assessment of plant nitrogen content [157]. Although UAV-based hyperspectral data provide information well-suited for phenotyping, data processing and analysis are often complex, requiring careful image calibration and pre-processing [158,159] to achieve spectrally accurate reflectance data suitable for scientific research and consistent monitoring [160]. Ivushkin et al., (2019) [161] used data from UAV-based multispectral, hyperspectral, thermal, and LiDAR sensors to discriminate between quinoa plants in a salt-treated plot and a control plot and found hyperspectral vegetation indices to be better suited than multispectral data for the spectral discrimination of quinoa plants between the two treatments. The LiDAR data were used to detect a lower plant height of salt-treated plants compared with the control plants. Temperature measurements could clearly discriminate the quinoa plants in the two treatments, when the thermal data were clustered based on plants with similar vegetation index values. These findings emphasize the potential of UAV-based phenotyping of quinoa plants. 

The successful application of UAV-based sensing technologies for phenotyping in quinoa field trials depends on the seeding pattern and dimensions because single rows of quinoa plants with limited spacing between rows (Figure 2A) precludes separation of plots owing to lodging across neighboring plots. Shading from tall quinoa plants of neighboring shorter plants affects their spectral reflectance characteristics as observed from RGB, multispectral, and hyperspectral imagery and plant temperature measured by thermal infrared cameras. To reduce shadow effects, it is generally recommended to collect RGB, multispectral, and hyperspectral UAV imagery close to solar noon. Although rectangular and square plots of greater width (Figure 2B,C) improve the separation of quinoa plants from neighboring plots for image analysis, except along their perimeter, the larger the plots and the greater the separation (Figure 2D), the more feasible it becomes to derive image-based representative samples of individual plots for analysis of phenotypic traits.

The collection of UAV imagery throughout the growing season allows multi-temporal assessment to study plant growth, behavior, and phenology. Multi-temporal assessment generally requires accurate georeferencing of the image datasets acquired during the growing season, which can be achieved using GPS-surveyed ground control points [140]. Variations in solar elevation, irradiance, and atmospheric conditions alter the illumination conditions over time. To enable spectral characteristics to be compared among multiple datasets, normalization of the image digital numbers to a set standard, normally at-surface reflectance, is required [162]. For consistent multi-temporal results and to ensure similar spatial resolution of the imagery, it is also recommended to employ the same flight pattern, altitude, and type of camera because different cameras are sensitive to different spectral wavelengths [140]. Finally, it is important to collect field calibration and validation data of phenotypic traits to be mapped. While in situ calibration data allow relationships and models to be developed, e.g., for UAV-based image classification, independent validation data enable the accuracy of maps to be assessed [163]. 

The use of automated and active phenotyping systems using light-induced fluorescence transient (LIFT) canopy scans can be useful for linking of photosynthetic performance and canopy structure and offers great potential for plant breeding and crop growth modeling [164,165]. UAVs, automated greenhouse phenotyping facilities, and rapid handheld phenotyping devices can create time series of imagery of large populations of plants over the course of a season or developmental stage(s) [166]. The data collected from these approaches may include a range of imaging modalities, e.g., hyperspectral, thermal, RGB, LiDAR, etc. This results in large volumes of high dimensional data for each plant that also contain spatial and temporal components. These data inherently contain cryptic information about biochemical, physiological, and morphological information of plants and their variance over time, and their conditionality on environmental conditions and genotype. Data volumes from such approaches can reach tens of terabytes per day and thus require automated approaches for phenotype extraction. Machine learning approaches, including deep learning, can be trained in a supervised manner to recognize phenotypes of interest such as height, chlorophyl content, flowering, plant architecture, abiotic stress, pathogen detection, disease quantification, etc. [167]. This is typically achieved by researchers manually labelling datasets and training neural networks to be able to predict those labels from the image data. After adequate training, the resulting neural network models can be used to analyze large volumes of data to extract phenotypes in an automated manner. However, the training phase is labor-intensive and supervised learning approaches will not, by themselves, extract all meaningful information from these complex, high-dimension datasets. As an alternative, unsupervised approaches such as autoencoders can be used to learn the latent spaces in a dataset and the resulting patterns extracted for use as phenotypes [168]. Traditionally, the downside of such an unsupervised approach has been the “black box” nature of neural networks which makes the extracted latent spaces difficult to interpret. However, recent advances in neural network architectures such as transformers with attention mechanisms show promise in relating the latent-space topology to the original features and thus producing interpretable phenotypes that are automatically extracted out from large datasets [169,170].

## 7. Phenotyping of Mature Plants

Toward the end of plant growth, before physiological maturity and harvest, it is useful to phenotype several traits at once in one large phenotyping event. Phenotyping at this stage can be seen as the cumulative effect of different developmental phases on the trait studied. Therefore, phenotyping at maturity can be a way to summarize the morphological strategy of the plant.

For most traits, the most useful stage for undertaking this intensive phenotyping is around mid-seed filling, during principal growth stage 8, i.e., ripening. It is important not to schedule this event too early because colors may not have developed or too late because during senescence, colors and leaves are lost. Depending on the number of traits decided upon for investigation and the size of the trial, this process is usually completed in 2–3 weeks. Preparation for the phenotyping event includes the set-up of a spreadsheet containing the field plan, plot numbers and associated accession information, and the traits that are to be described. An example of such a table lies in the templates linked on the home page of Quinoa Germinate Database. Available online: http://germinate.quinoadb.org (accessed on 15 August 2021). Not all phenotypes will be recorded in each trial; only phenotypes that show variation across the field should be selected. In addition to the descriptions, phenotyping cards are created (Appendix A). These may be printed, laminated, and given to each person of the phenotyping team to carry while assessing plots within the field. 

Multiple teams of individuals may divide the workload to shorten the time required for completing the phenotyping. However, this comes at a risk of introducing biases and errors. If the work is shared, it is crucial to train each person adequately and to ensure that all agree with the method using which traits are scored or measured. Hence, it is important that extra time for training each person is planned in for the first day of phenotyping. The first plots are scored by all people together. This gives an opportunity for ensuring that the scoring method for each phenotype is clear to each person. Next, around five plots should be scored independently by each person, and scores should be compared. Where differences in the given scores are identified, these should be discussed. The next plots can only be split up between the people scoring, once all are in agreement. At multiple times throughout each day in the field, each team should phenotype overlapping plots and compare the measurements to ensure that the measurements agree and traits persist. The following section describes in detail the phenotypes that we recommend to measure during phenotyping of mature plants. 

### 7.1. Assessing the Quality of Phenotypic Data

To address questions that may arise during the analysis after termination of the trial, for instance, checking the data for outliers or apparent typos, photographs are taken of each plot. Four photographs are taken, as shown in the example images in Figure 3, starting with one picture of the entire plot. It is useful to have a pole used for height measurements included in the middle of the plot as a scale reference. Next, one representative and easily accessible plant (avoiding the outer edge plants) is chosen for a second picture, which should show the entire plant. For this picture and the following close-up picture of the panicle, a black background is needed to capture all details of the individual plant. This backdrop should be composed of cloth with a matte finish to prevent light reflectance and should be attached to a wooden frame that can be easily transported. A color rendition chart (e.g., ColorChecker Passport Photo 2, X-Rite Inc., Grand Rapids, MI, USA) is also important, to include color calibration because light conditions may vary highly throughout the day depending on the weather conditions. Therefore, a camera with high resolution and manual adjustments of the settings to permit adaptations to changing light conditions is needed. For example, the images in Figure 3B,C were taken with a Canon EOS 70D device (Canon, Tokyo, Japan), with manual settings for F-number (f/7.1) and exposure time (1/250). However, for the plot image, this exposure time resulted in an overexposed image (too bright). A second photograph was taken with exposure time 1/500 (Figure 3A). Different settings were used again for Figure 3D, the image of seeds on the blue card. This photograph was taken using the automatic setting “aperture priority” instead of a manual setting, and used a different F-number (f/9). For image analysis, images need to be stored either as raw data files or using lossless compression techniques to enable complete reconstruction of the data from compressed data. This is also important because the quality of the images taken determines whether they can later be used for image feature extraction. Algorithms for the extraction of plant morphological features from these types of images are currently being developed, and seed phenotypic characteristics can already be extracted from seed images by placing the seeds on a blue card background (Figure 3D). This technique is described in detail later (Section: Seed Phenotyping, Section 9.2.1.). 

### 7.2. Plot-Level Phenotypes

In addition to the photographs of plants, there are a number of phenotypes that should be scored to allow others to do quantitative analyses without the need to be involved in data collection. A number of quality control phenotypes are assessed as the percentage of the plot that is affected (Table 2). These phenotypes may also need to be considered in analyses of the trial data where the phenotypes might affect any of the later described “plant-level” phenotyping data in which one representative plant for the plot is chosen for further measurement.

#### 7.2.1. Plot Population Homogeneity

The most important trait for quality control of data in the analyses of datasets is the score of heterogeneity of the phenotypes of the individual plants within a plot. The genetic diversity of quinoa is wide owing to less-intensive breeding events (and thus a relative paucity of population bottlenecks), and several quinoa accessions are landraces that produce a heterozygous phenotype. Quinoa is predominantly self-pollinating and has varying rates of natural hybridization of 10–17%, which are likely to be greater at lower plant spacings, and depending on the coincidence of flowering with the windiness of the site or the presence of other pollen vectors [7,171]. There is also a possibility for outcrossing if panicles are not isolated with a bag. The heterogeneity of a population can be beneficial in small-scale cultivation, where it might confer greater yield stability in unpredictable weather conditions. However, for genetic studies, heterogeneity poses considerable challenges because the phenotype must be correlated with the genotypic information. Hence, highly heterogeneous genotypes are not suitable for genetic studies, and plots need to be excluded from the analysis if >50% of the plants within a plot are segregating, i.e., they are observably different, and a plot must be excluded when the main phenotype of the accession is not identifiable within the plot (for example images of the different categories of heterogeneity scores, see Appendix A). If possible, producing inbred genotypes by isolating the panicles within 100 µm mesh pollination bags (“bagging”) before the start of genetic studies is recommended. When bagging panicles in the field, bag size is important. Bags of size 10 × 15 cm were found to be best; larger bags can be caught more easily by the wind. After flowering (at BBCH 70), the bags should be removed to allow the panicles to expand and grow. It is important that the panicles remain tagged after the removal of the bags to keep information on which panicle is to be harvested for pure seeds. For example, bags can be tied to the plant underneath the panicle upon removal.

Heterogeneity is scored by assessing the percentage of plants in the plot that have a visibly distinct phenotype from the majority of plants in the plot, where four categories are described as follows (see also Appendix A): 1: Most plants are the same (up to 10% different).3: Over half of plants are the same (10–30% different).5: Less than half of plants are the same (30–50% different).7: Over 50% of the plants are different, completely mixed plot; will need to be excluded from analysis.

#### 7.2.2. Plot Coverage

Another quality control measure is plot coverage, which can give an indication of the spatial heterogeneity at the plot level. The importance of recording plot coverage lies in the aforementioned phenotypic plasticity of most quinoa accessions in response to the space available around them, which is why this information may be included as a covariate in analyses of responsive traits. 

The plot is assessed using a scoring metric based on percentages, where:1: Up to 20% of the plot is covered, plant establishment is very poor.3: Less than half of the plot is covered, ~30% (20–40%).5: Around half of the plot is covered, ~50% (40–60%).7: Over half of the plot is covered, ~70% (60–80%).9: Over 80% of the plot is covered, plant establishment is very good.

To make the assessment of plot coverage in percentage easier, and if the plot size allows it, plants can be counted to see how the number compared to the total amount of plants a plot should have. Alternatively, plot coverage information can be deduced from plant emergence or UAV data, if available.

#### 7.2.3. Stem Breakage Incidence

Other factors such as plant damage should also be considered in subsequent analyses of traits. For instance, damage to the stems may be relevant for yield measurements when the damage causes a loss of yield from affected plants. Stem breaks are often caused by strong winds, but can also happen after insect damage or some fungal diseases (which should be checked when stem breakages are observed). Because panicles get detached from the plant, further progression of their life cycle is prevented. Depending on the timing of the damage event, the affected panicles may still be harvestable. Stem breakage is an undesirable trait in a cultivar and is assessed on the basis of the percentage of the plot affected, where: 1: Up to 20% of the plot is affected.3: Up to half of the plot is affected, ~30% (20–40%).5: Around half of the plot is affected, ~50% (40–60%).7: Over half of the plot is affected, ~70% (60–80%).9: Over 80% of the plot is affected.

#### 7.2.4. Stem Lodging and Stem Angle

In contrast to the snapping of stems, stem lodging refers to bent plants lying on or near the ground, with intact stems. Although the panicles are still harvestable in this case, depending on the degree to which a plant lodges and how close it is to the ground the panicle is, they might not be picked up by a combine harvester. In addition, where panicles are lying on the ground, pre-harvest sprouting and fungal infections may arise. Often the proportion of the plot that is affected by entirely lodged plants varies. Hence, for scoring stem lodging, the percentage of the plot affected is recorded using the same percentage-based categories as described above. 

Alternatively, stem vertical angle, i.e., the angle at which the majority of plants in the plot are leaning towards, measured from the vertical axis, may be scored (see Appendix A). Here, a scoring system is used, where: 1 < 22.5° inclination or deviation of the stem from the vertical (i.e., most plants are upright).3 < 45°.5 < 67.5°.7 < 90° (i.e., most plants are on or very close to the ground).

#### 7.2.5. Panicle Axis Angle

Depending on the environment, or if a treatment (for example salinity) is applied, genotypes can be observed to have panicles that droop towards the ground. A deviation from the vertical may also represent a heat avoidance strategy as was observed in sunflower [172]. Similar to stem vertical angle, panicle drooping is assessed for the majority of the plot and based on the degree at which a panicle deviates from the vertical. Because a panicle can droop towards the ground while its stem remains vertical, this scoring system goes up to 180° in this case:1 < 45° inclination or deviation of the panicle from the vertical (i.e., most panicles are upright).3 < 90°.5 < 135°.7 < 180° (i.e., most panicles are pointing towards the ground).

#### 7.2.6. Stem Lying Incidence

Stem lying sometimes co-occurs with stem lodging. The cause of stem lying remains unclear, but it results in the lying of stems on the ground, which may occur at the seedling stage of emergence (BBCH 00–09). The length of the section of a stem that is growing along the ground varies, indicating that the plants with stem lying could be classified into distinct groups. However, during in-field experiments, with multiple plants in a plot, it is apparent that there are some complications with this phenotype. Some plants exhibit severe stem lying, whereas others with the same genotype in the same plot have none. Hence, when stem lying is noted in the plot, the percentage of the plot that is affected should be scored, not the severity of the lying itself. The scoring categories are therefore the same as those for stem breakage listed above. Severely affected plots may need to be excluded from subsequent analyses. Depending on the severity of this phenotype observed in the field, stem lying may not be necessary to record. 

#### 7.2.7. Growth Habit

The architecture of quinoa plants varies greatly, and because this phenotype is particularly responsive to environmental conditions, the growth habit category for an accession is also classified at the plot level. Depending on the experiment, growth habit may also be assessed for individual plants. 

The categories shown in Figure 4 were drawn based on their presentation in the Quinoa Descriptors [38], but the description of the categories was adapted to include the extent of growth habits that we observed in our diversity panel of approximately 1000 accessions. The feature that differentiates the groups is branching in the lower-third of the plant as well as the size of the panicles on those branches.

1: Not branched at base, usually with a clearly defined terminal panicle.3: Some branching from the base; no significant panicles on branches in the basal area (thus, this is not worth harvesting).5: Branching from the base with more significant panicles.7: Main panicle is difficult to identify.

#### 7.2.8. Branchiness

Growth habit is focused on the branching habit at the base of the plant; however, in this category, we assess the degree of branching across the entire plant. The number of branches coming from the primary axis is easiest to assess when the plant is in principal growth stage 9, senescence. Because the phenotyping event is planned for a time where most plants are at principal growth stage 8, attention should be paid to avoid bias from the leafiness of the plant when scoring for branching degree. Plants from the middle of the plot are categorized into:1: Low number or no secondary branches.3: Some branches (30–50% of the primary branch length has secondary branching).5: Branched (50–70% of the primary branch length has secondary branching).7: Highly branched (above 70% of the primary branch length has secondary branching).

### 7.3. Plant-Level Phenotypes

After assessment of plot-level phenotypes, it is recommended that detailed observations of representative plants from each plot are made. As representative plants, again we select individuals from the middle of the plot (to avoid edge effects) which share features with plants across the entire plot, including the traits outlined in Table 3 and Table 4. Depending on the time available for plant phenotyping, we select 1 to 3 individuals per plot. If there is a clear segregation of phenotypes in a plot, these plots are to be marked heterogeneous and are excluded from genetic studies. Representative individuals from each “type” can be separately phenotyped to maintain a record of how the accession segregated phenotypically. The phenotypes measured for each plant are divided into quantitative and categorical traits, as outlined in the following sections.

#### 7.3.1. Quantitative Plant-Level Phenotypes

An overview of the quantitative traits is shown in Table 3. They are presented based on the order used during phenotyping in the field. 

##### Plant Height

Plant height is measured from the base of the plant at soil level to the tip of the primary panicle. We found that the use of metal ranging poles with alternating red and white colors to be the best option for a height reference because they are sturdy, yet not too heavy, metal poles with alternating red and white colors which work as height references. Extra markings and height labels need to be added to the poles using black tape and permanent markers. As ranging poles are made up of multiple parts that are screwed together, the height of the poles can be adjusted to the maximum height of the plant in the trial. If representative plants are not entirely upright, they can be held upright to record their height.

##### Panicle Length

Panicle length is measured for the primary panicle from the base of the panicle to its tip. In most commercial varieties, panicles are often extremely distinct and easily measured. However, in some quinoa accessions with wild phenotypes resembling *Chenopodium hircinum* accessions, panicles can be sparse, with the inflorescences spaced along the length of each branch. In these cases, we grasp the panicle below the lowest lateral branch with a large inflorescence and bring them together to get an estimate of the primary panicle. 

##### Stem Diameter

Stem thickness is measured twice, once at the base of the plant and once just below the panicle. In our protocol, digital calipers are used to measure both. In dense plots or with large field trials, these measurements can be very time-consuming and are best applied for studies assessing the likelihood of stem lying, stem-breaking, and lodging. 

##### Number of Significant Panicles

An attractive trait for a commercial variety that is harvested by machine is the presence of a single primary panicle. However, in many accessions, several additional panicles can be observed emerging from lateral branches. A significant panicle is a panicle which is large enough to make an important contribution to the yield obtained from a plant. These panicles are often on the upper half of the plant, near the primary panicle. The number of significant panicles should be counted, which is generally equivalent to the number of primary branches. The “significant panicles” should, together, contribute to an estimated 90% or more of total plant yield.

##### Categorical Plant-Level Phenotypes

Aside from the phenotypes that are measured, a number of traits are visually assessed and assigned to defined categories. An overview of the qualitative traits is shown in Table 4.

##### Seed Shattering

Dehiscence (or seed shattering) is a dispersal strategy of importance to wild plants, but this trait is a major cause for crop yield losses [173]. Increased persistence of the grain within the panicle is therefore a priority trait during the domestication process of a crop. This trait is assessed by lightly tapping 3–5 panicles while holding the other hand or a piece of paper underneath the panicle to catch the seed that falls off. The number of seeds that fell may be:1: No seeds falling.3: Some seeds falling.5: Many seeds falling.7: Majority of seeds is falling, “raining” seeds, and a large number of seeds present on the ground at measurement.

##### Panicle Shape

Three categories have been described for the overall shape of quinoa panicles [38], as illustrated in Figure 5. However, this trait has caused problems, as some panicles are not easily categorized into the described groups. The three groups were changed to scores of 1,3,5 instead of the scores previously named 1,2,3 in the Quinoa Descriptors [38] because quinoa panicles are so diverse that panicles cover a wide spectrum of shapes, rather than falling into the three distinct groups that are described below, or the two categories that were previously suggested, glomerulate and amarantiform [123]. The difficulty in categorizing is causing inconsistencies with scoring. Glomerulate panicles usually have clusters of glomerules at the end of a cluster of branches emerging from the secondary axis, as shown in Figure 6. To distinguish between the intermediate and amarantiform groups, the length of the secondary axis, which is usually packed tightly with glomerules up to the junction (resembling “fingers”), should be considered. Additionally, intermediate panicles can have short tertiary branches emerging from the secondary axes, usually from the bottom half of a “finger”. The glomerules inserted into the short tertiary axes create “bulbous clusters” and lead to the presence of both, glomerulate and amarantiform features in an intermediate trait. Generally, it was also observed that panicles fitting into the intermediate category have their elongated glomerules only starting from a lower half of the panicle, while the top resembles a triangular shape of glomerulate glomerules. To improve this classification system in the future, image analysis algorithms are being developed for the identification of new groups from images of panicle inflorescences. 

1: Glomerulate—glomerules with globose shape, resembling “bulbous clusters”.3: Intermediate—panicles have both amarantiform and glomerulate traits, resembling fingers with glomerules.5: Amarantiform—glomerules with elongated shape, resembling “fingers”.

##### Panicle Density

Panicle density, a trait that can vary greatly with environment (such as temperature), contributes to the complications with defining panicle shape and is scored separately as follows:1: Lax (loose)—glomerules sparsely spaced, panicle axes easily visible.3: Intermediate—glomerules tighter but with panicle axes still visible.5: Primary axis rarely visible.7: Compact—glomerules tightly packed, no panicle axes visible.

##### Panicle Leafiness

Within the inflorescences, there are often leaves growing among the flowers (see Appendix A). Variability between genotypes and environments is observed for this trait and can be scored as follows: 1: Leaves are present in less than one-third of the panicles.3: Leaves are present in more than one-third but less than three-fourths of the primary, sporadic, and not dense panicles.5: Leaves present in three-fourths to of the entire primary axis, frequent but not dense leafiness.7: Many leaves present throughout the primary axis.

##### Panicle Color

Color code descriptors with 15 colors for quinoa panicle scoring were previously provided by [38]; however, scoring for color is highly subjective. We find disagreement among individuals scoring panicle colors, particularly for differentiating red, pink, and purple. This problem highlights the need for providing color cards (e.g., Royal Horticultural Society Colour Chart, Methuen Handbook of Colour, or Munsell Color Chart for Plant Tissues) as a direct reference when accurate classification of colors is of interest, and in this case, it is advisable to extract color information from images. In most field trials, detailed color recording may not be a priority, but recording the most prevalent color is a useful indicator of phenotypic segregation and quality control as color is a dominant morphological marker [174]. Therefore, we reduced the number of categories to the following six:Green (13);Green with Purple (16);Pink/Purple/Red (4);Orange/Yellow (5);Dark colored (7);Beige/White (i.e., no pigmentation, mostly for mature plants) (15).

##### Stem Color

For stems, 11 colors were proposed [38]. This wide color variation has not been observed in the field among our diversity panel and is difficult to correctly identify under field conditions. Stem color is useful to serve as a quality control for validating protocols in genetic analyses because the loci associated with stem color have already been identified [15]. Therefore, plants only need to be categorized into the following:Green (13);Red (4);No pigmentation (beige, white, yellow) (15).

Stem color can also provide information about the homozygosity of an accession, and accessions have been observed with red coloration at the leaf–stem intersection, which may be used as a control for whether crosses have been successful [174]. A lack of pigmentation is based on panicles and stems that have lost their color owing to senescence. It is not always possible to time the scoring campaign for color across all accessions because there can be great variability in the number of days required to achieve maturity.

##### Stem Striae and Axil Pigmentation

The presence of stronger pigmentation forming stripes on the stems and pigmented axils are traits that persist and may be used for example for the identification of successful crosses. If desired, stem striae color can also be recorded. Separate genetic mechanisms to color may likely regulate the location of the pigment accumulation or synthesis. Hence, it should be sufficient to score for: Presence (1);Absence (0).

##### Stem Leaf Shape Characteristics

Leaf shape characteristics are heritable traits that may be useful to breeders for variety identification. These traits may also be correlated with irradiation capture, water use stability or yield stability. They need to be recorded at flowering. As described by [38] stem leaf shape can be categorized into two groups: Rhomboidal (1);Triangular (2).

The leaf margin refers to the shape of the edges of the leaf. This trait also summarizes teeth number, another leaf shape characteristic which can be scored separately if more detail is required. Leaf margin is categorized into:Entire (1);Dentate (3);Serrate (5).

## 8. Phenotyping of Disease

Depending on the environment in which a trial is conducted, quinoa may also be affected by diseases that influence plant health and yield negatively. Therefore, recording of disease occurrence and scoring for disease severity might be necessary in field trials where diseases are observed despite taking pest and disease control measures. As infections can progress rapidly, it is important that the plots of a trial are regularly assessed for disease throughout the growing season, and controlled for [175]. Trials where disease is not controlled but representing the studied treatment are independent trials designed for the phenotyping of disease, typically by the selection of genetically diverse accessions with variable disease resistance. Disease assessment includes measuring the incidence (the number of affected plants out of the total assessed) and severity (proportion of plant area or fruit volume destroyed by a pathogen) [176]. Observations must be conducted at least three times: once during phenological growth stage 1, when plants have around nine true leaves, and before branches develop; a second time during either principal development 4 (development of harvestable vegetative parts), principal growth stage 5 (inflorescence is visible), or principal growth stage 6 (flowering); and a third time during principal growth stage 8. To avoid difficulties arising with senescing leaves, observations should be conducted before principal growth stage 9. 

Accurate diagnostics of quinoa diseases, however, are complicated because multiple pathogens often appear in communities [177]. A field situation is a complex interaction between the plant and its microbiome. Therefore, incorrect identification of the pathogen involved can occur if only a single organism is considered. Some plant disease agents can be identified through their symptoms and classified from infected tissue by skilled plant pathologists. However, numerous pathogens cannot be distinguished from each other based on the visual assessment of disease symptoms. In fact, molecular tools and clear distinctions among quinoa plant diseases were lacking in the past, with only a few examples properly described. The pathogens identified to affect quinoa include the fungal pathogens *Ascochyta caulina, Cercospora* cf. *chenopodii, Colletotrichum nigrum, C. truncatum, Fusarium* spp., and *Pseudomonas syringae* [175]. The predominant and most well-described pathogen is the oomycete *Peronospora variabilis:* it causes downy mildew, and its impact is considered to be one of the most economically important [51,178,179,180,181,182,183,184,185,186]. The complexity of diseases can lead to inaccurate diagnosis in field trials. Therefore, we propose for the in-field phenotyping of disease symptoms to assess all three parts of the plant: foliage, stem, and panicles. Leaf symptoms should be well-described with respect to detectable changes in lesion shape (irregular, blotch, spots) (Figure 7(Aa,b); 7(Ca,b), color (pink, bronze, chlorotic, mix) (Figure 7(Aa–d), other symptoms (surrounding halos, concentric rings) (Figure 7(Aa,d)), and distinctive signs such as sporulation usually present on the abaxial side of the leaf (Figure 7(Ba,b)), and chlorotic leaf veins (Figure 7(Bc)). Next, the type of lesions on a leaf (dots, diffuse or extensive) are recorded (Figure 7(Ca–d)), followed by the amount of spread which is measured by the percentage of leaf area affected (Figure 7(D1–5)) in relation to its total area. The scoring could be based on a scale from 1 to 5 (1 = 0–10%; 2 = 11–25%; 3 = 26–50%; 4 = 51–70%; 5 = 71–100%), which is frequently used as a measurement of disease caused by various pathogens including severity of downy mildew of quinoa [187].

For the estimation of severity in the field, we suggest a selection of 3–10 representative leaves. Disease symptoms are often influenced by the age of the plant and position of the tissue [176,188]. Leaves at the base of the plant could be displaying symptoms of senescence. In contrast, leaves toward the apex of the plant may display induced resistance which often occurs not only at the site of the initial infection but also in distal uninfected parts [189,190]. Therefore, we propose that samples selected from the middle part of the plants best represent the infection. In the case of stems and panicles, similar principles should be applied. Examples for visible changes in panicles and stems brought upon by pathogens are shown in Figure 8. Panicles and seeds can be affected by various pathogens causing different amounts of rotting, which can result in total harvest loss and contaminated seed. The example for stem disease in Figure 8B shows a mature stem with necrosis of diamond shape (which can cause plant lodging), concentric rings, and visible pycnidia. Another young stem shows pink coloration and white mycelia. 

For accurate disease diagnostic, further procedures are needed: samples should be saved for isolation, microscopic analysis, and molecular identification of the causal agent. Koch’s postulates, guidelines for determining a causative relationship between a microorganism and a disease, should be validated. The steps required for the standardized procedure include (a) description of disease symptoms, (b) isolation of the disease agent, (c) artificial inoculation of quinoa tissue with the isolated agent, (d) recording of symptoms on the infected quinoa tissue followed by (e) a re-isolation of the microorganism from the infected tissue [133,176,191]. More ongoing research is required to identify possible pathogens, especially newly emerging ones, behind quinoa diseases. Advancement of molecular diagnostics such as development of rapid DNA extractions and newly designed species-specific primers along with advanced remote sensor-based image techniques are expected to be helpful for fast and more accurate disease detection in the near future. 

## 9. Harvest and Post-Harvest

When all other phenotypic traits have been collected and plants have reached maturity (BBCH 89), they may be harvested and prepared for post-harvest data collection. Depending on the trial size, type, and resources available, harvest may be approached using different methodologies. Irrespective of the method selected, it is important that an indication of the number of plants that contributed to the yield is available, by counting the plants when hand-harvesting, harvesting only a set number of plants per plot, or harvesting the entire plot and using previous field emergence information. Recording the plant number allows the yields obtained in one trial to be compared with results from another trial. This is crucial information to collect for international collaborative quinoa research. The recommended harvest and post-harvest traits are listed in Table 5.

### 9.1. Harvest Protocols

When developing harvest protocols, it is important to consider edge effects. Therefore, when choosing a plant for phenotyping, it is best to select only plants from the inside of the plot, while disregarding unusually small plants or the typically larger and more branched individuals at the borders, which might cause a bias in the yield predictions. Similarly, in plots with uneven emergence, unusually large individuals should not be selected. Depending on the plot size, it is recommended to harvest a larger number of representative individuals (20–30 plants) because smaller sample sizes may lead to less accurate predictions for yield. The number of harvested plants as well as the plot coverage should always be noted. Main panicles may be harvested separately from secondary panicles of a plant to obtain an indication about the distribution of seed on a plant. 

For calculation of the harvest index, i.e., the ratio of harvested seeds to total dry above-ground biomass, a subset of plants should be selected for harvest as entire plants and cut at the base of the plant at soil level using secateurs. Depending on the plot size, 4–6 plants may be used for this evaluation. The plants should be placed in bags and dried in in an oven at no less than 60 ℃ for several days (until weight is constant). Once plants are dry, they can be weighed to obtain shoot biomass before proceeding to threshing and weighing the seed. This information is extremely important for identifying the best-performing genotypes, which are those that invest more resources in their seeds rather than to their above-ground biomass.

Threshing and winnowing are, together, the process of separating the seed from the chaff or straw and is easiest when the panicle is dry. Before mechanically threshing, the seed should be loosened from panicles and chaff by hand to facilitate the threshing and winnowing processes and reduce seed loss during machine threshing. After threshing, the seeds are weighed to obtain yield. A sample can be taken and weighed to obtain the Thousand Grain Weight (g/1000 seeds). However, for higher throughput and more detailed information, the seed scanning method described in the following section is recommended. 

### 9.2. Seed Phenotyping

Because the seed is the final product, seed properties, especially their nutritional properties, are important to be considered when selecting varieties of interest. Seed color was not found to be correlated with significant differences in most nutritional properties, except perhaps protein and carbohydrate contents. Pereira et al., (2019) [192] have reported that white seeds had the lowest protein and highest carbohydrate contents compared with red and black seeds. However, color is of interest because large white seeds are in demand in the market for quinoa. In some countries, such as Bolivia and Peru, large red or large black seeds are also desirable. Irrespective of the color, large grain size is the highest priority for international market [7].

The phenotypes of seed color and grain shape were previously divided into different categories [38]. However, visual assessment of the seed and assigning it one of four not easily distinguishable categories for shape is not very precise. With color identified as a highly subjective trait, in the following section, we present an alternative that saves considerable time spent on phenotyping these traits without the need for expensive equipment.

#### 9.2.1. In-Field Seed Morphology Descriptors

An image feature extraction algorithm was developed to produce the following seed morphological descriptors from an image: seed area, perimeter, color, and counts. Although centralized image collection with controlled lighting and fixed resolution provides the highest quality images, the collection of images in the field can save time and resources. This algorithm also provided accurate results on images that were obtained using the camera of a mobile phone. However, adjustments to the feature extraction methods are required because images from these cameras are distorted. 

Apart from a camera, a backdrop for quinoa seeds is needed. A blue card of fixed size is best. The blue color provides a color contrast with the quinoa seeds, and the fixed size allows pixel to mm conversion. A template for this 10 × 10 cm^2^ square blue card is provided (Appendix A) and should be cut from weighted paper of the reference color. Overall, 10–50 seeds are placed on the card, and a photograph is taken. In the case of in-field image collection, the image may be taken with a camera held in one hand, while the card with the seeds is held in the other, as shown in Figure 3D.

The measurement of seeds on the blue card comprises three steps: isolation of the blue card from the scene, rectification of the card to a square of known pixel count, and segmentation and measurement of the seeds from the card. The image analysis software CyVerse was made available to the community on CyVerse [193]. In the CyVerse Discovery Environment, images can be analyzed using the Phytomorph Image Phenomics Toolkit. After clicking on the app, an analysis name can be assigned and single images or folders that are uploaded to the CyVerse Data Environment can be selected for analysis. For this, the required image analysis algorithm must be selected. The algorithm for the method described here is called “Quinoa Seed Card”. Once the analysis is completed, the user is notified via email and can return to the Discovery Environment where the outputs are stored in the user’s database. The outputs can be browsed there or downloaded. The results include values for average area, average perimeter, and average red, average green, and average blue components. In addition, images are returned with seeds that were detected by the algorithm as single seeds, used for analysis, and highlighted in red. This allows confirmation that the algorithm showed expected performance. 

#### 9.2.2. Seed Scanning

For more accurate measurements of seed morphological characteristics, including size, shape, and color, a high-throughput seed scanning system has been established in the Sustainable Seed Systems Laboratory (SSSL) at Washington State University (WSU) to capture images of quinoa seeds. Similar systems may be established elsewhere; however, users must consider their needs. The SSSL system uses eight flatbed scanners, which capture images at a resolution of 1200 dots per inch (dpi). System design is focused on the ability to queue and initiate four samples on one set of scanners, with the process being repeated on other set of scanners while the first set of images are captured. This system allows the analysis of approximately 50 samples per hour and supports data collection for thousands of samples in a year. 

Users should consider their desired throughput when designing a system to best meet their needs because this will determine the number of scanners required and their orientation in the workspace. Certain pieces of equipment are required, regardless of throughput (Appendix A). The SSSL system uses Epson Perfection V39 flatbed scanners (Epson America, Inc., Long Beach, CA 90806), which balance affordability (USD 50–100) with capability (up to 4800 dpi optical resolution; 600 dpi color scan in 30 s). A small subsample (approximately 1–2 g) of each sample is taken for seed scanning. A sample splitter or other comparable equipment should be used to collect a representative subsample. Seed samples are weighed before scanning; however, scanners with integrated balances are available. The SSSL uses an Ohaus Scout SPX123 Analytical Balance (Parsippany, NJ, USA), with a capacity of 120 g and a precision of 0.001 g. Seeds are carefully scattered on the scanner, within the field of view, to limit the number of seeds that are touching. Then, the scanner lid is closed. The underside of the scanner lid, acting as the image background, has been painted using the flat, matte paint color “Blueberry Festival”, which can be characterized by red, green, and blue values of 73, 139, and 184, respectively, and HEX #498BB8 (Valspar Paints, Cleveland, OH, USA).

Scanners are connected to a Dell Optiplex 9010 SFF Computer (Intel Core i5-3470 3.2 GHz, 16GB RAM, 2 TB HDD) with USB cables. This machine runs a Linux operating system (Ubuntu 18.04) to support simultaneous scanner operation and image filing using Python shell scripts. The system is command-line driven, where QR codes are scanned using a wireless 1D/2D barcode scanner. The QR codes have embedded information related to the scan command, scanner identity, and sample identity to facilitate high-throughput operation and reduce the risk of human errors associated with manual information entry. Images captured on each scanner are saved, backed up on an external hard drive, and uploaded to the CyVerse Discovery Environment for image analysis.

Subsequent image analysis provides a robust data set for use in phenotyping and quantitative analyses. Image analysis is performed in the CyVerse Discovery Environment using the Phytomorph Image Phenomics Toolkit, the same application that was used for the analysis of the blue card images described previously. First, a single high-throughput file path is created with a set of images. Next, the high-throughput file path is selected for analysis with either the Arabidopsis Seed Method (black background) or All Grains (blue, lilac, green or white background) single file tool in the Phytomorph Image Phenomics Toolkit. A high-throughput multifile path is created from the output using .json files. This high-throughput multipath is then selected for analysis using the JSON compiler in the multifile tools in the Phytomorph Image Phenomics Toolkit. Finally, a .csv file of results for each image is available for download. The data include measurements of each seed in the image, with mean and standard deviations. A tutorial demonstrating this process is available on the SSSL YouTube channel, SSSL Training - Image Analysis. Available online: https://www.youtube.com/watch?v=9SK4vkfeJHI (accessed on 15 August 2021).

The CSV file from the seed image analysis contains a large amount of information. Seed size, shape, and color are quantified. The number of seeds in each subsample is automatically counted, and this information is combined with the weight of the subsample to calculate the weight of 1000 seeds (Figure 9). Seed size includes seed area and the length of the major and minor axis, which are used to calculate seed shape (i.e., eccentricity). Seed color includes red, green, and blue values that can be used to produce any color. Seed count is reported as the number of objects in the image. If seeds are touching, the size of that shape is divided by the average size seed to determine the number of seeds in the aggregate. Means and standard deviations are reported for each measurement. The mean values represent an average calculated using the values for each seed, with the standard deviation representing the variability within the sample. For some measurements, such as color, a standard deviation of standard deviations is reported; a standard deviation for the sample is calculated using the standard deviations of all the seeds in the sample (Appendix A).

#### 9.2.3. Seed Nutritional Phenotyping

##### Near-Infrared Spectroscopy

Quinoa seed composition, especially protein content and composition, can be highly variable [194,195]. Seed composition can vary depending on processing, when certain parts of the seed are removed or modified through washing, abrasion, or milling [196]. Therefore, seed composition analyses and their results must be considered in the context of processing and physical state of the quinoa seed before analysis. One method to achieve HT analysis of quinoa seed composition is the use of near-infrared (NIR) spectroscopy, in which, in theory, any sample measurement can be predicted as long as the sample’s spectral data are correlated with the desired measurement [197]. This technology has diverse capabilities, such as phenomic selection and prediction of maize yield from kernels [198], and has been successfully applied to predict amino acid content in quinoa [199]. In addition to the NIR methodology, mid-infrared should be considered as a useful technology with practical applications in quinoa. For example, mid-infrared has been used to classify groups of quinoa [200] and characterize rheological properties [201].

An NIR calibration requires careful selection of samples to serve as the reference data, from which multivariate regression equations will be created. Spectral data are used to predict the target measurement(s). Sample selection should represent the target population of spectra [197]. One approach is to collect spectral data for as many samples as possible. These samples should represent the genotypes and environments that will be routinely analyzed. Next, principal component analysis of the spectral data, combined with the Kennard–Stone method of sample selection, can identify the best candidates to include in the calibration [202]. The R package prospectr provides multiple options for selecting samples for calibration and validation sets using a multivariate spectral data set. Another approach is to select samples representative of a normal distribution for the desired measurement, such as protein content. 

Reference data, such as protein content and amino acid composition, should be measured according to Official Methods of Analysis [203]. Blind duplicates, i.e., multiple samples from the same seed sample submitted for analysis and acting as quality control, should be included when possible to estimate and account for errors during calibration development. Various methods can be used to develop calibration equations, such as partial-least squares regressions. The calibration metrics, such as standard error of prediction, should be quantified through either internal or external validation processes [198,204]. The development of calibration processes is ongoing. The initial calibration should be updated over time by incorporating outliers into the reference data set or additional products (market classes, flours, etc.) into the crop analysis profile to improve the calibration(s). 

##### NIR—An Example of Calibration Development for Quinoa

When beginning to develop an NIR calibration, one should consider how the quinoa samples will be processed before analysis and determine the desired outputs from the analysis. For example, the WSU SSSL has initiated NIR calibration development with raw whole quinoa seed, which are unprocessed but cleaned of non-seed material to predict crude protein, crude fat, ash, and moisture content, in addition to a complete amino acid profile using a DA7250 (PerkinElmer, Waltham, MA, USA) with an NIR range of 950–1650 nm. The advantage of this approach is that samples require minimal processing before analysis. The disadvantage is that sample homogenization through milling could provide a more accurate representation of seed composition because the entire seed is made available for spectral reflectance. Scatter corrections, such as standard normal variate (SNV) or multiplicative scatter correction (MSV) can be applied during pre-processing of spectral data to correct for differences in the sample matrix (i.e., seed surface) [205]. A non-destructive approach ensures that seed viability is maintained to support subsequent breeding activities, such as greenhouse seed increases, field trials, and further research and analyses, such as food science studies. 

The SSSL has improved the stock DA7250 NIR calibration twice. The stock calibration included 27 samples and reference data for moisture, protein, and ash. However, it was not robust and poorly predicted novel quinoa samples; NIR analysis is best at interpolation rather than extrapolation. The second version (V2) added samples from the WSU breeding program materials and field research trials [206]. These samples were randomly selected across a normal distribution of crude protein content predicted using the stock calibration. The current NIR calibration (V3) incorporated 37 samples selected from a collection of diverse genotypes grown in Australia in 2018. Samples were selected using the Kennard–Stone method, with principal component analysis of the spectral data. The reference data collection included 10 blind duplicates to measure and account for laboratory standard error in the calibration.

Eight-fold cross validation was performed in triplicate to measure calibration prediction accuracy metrics, which are reported as an average measure (Table 6). These metrics provide an indication of how well the calibration may perform. For example, a large range in reference data values is beneficial for encompassing the possible values of experimental samples that may be analyzed, and usually contributes to higher prediction accuracy as measured by the correlation coefficient between measured and predicted values. The metrics related to the cross validation provide various measures of the calibration prediction accuracy. Although the current calibration includes various seed colors, calibrations specific to the major quinoa seed colors—white, black, and red—may be more appropriate. NIR calibrations exist for developed market classes in other crops, such as red and white wheat. The SSSL will continue to analyze diverse genotypes grown in varying environments for identifying candidate samples with the potential to improve the accuracy of the NIR calibration for predicting quinoa seed composition. This will be achieved by either including samples with novel spectral signatures, or by including data that increase the range of reference data for particular seed components.

##### The Nutritional Phenotyping Pipeline at Washington State University

HT analysis of seed composition and characteristics is performed by SSSL at WSU using a Nutritional Phenotyping Pipeline. The system has been applied to barley, perennial grains, quinoa, and camelina and has the potential to be applied to diverse crops. The system is flexible and can accommodate varying sample amounts. Seed characteristics can be measured using image analysis with as little as 1–2 g of quinoa. With approximately 10 g of quinoa, seed composition can be estimated using a NIR analyzer; however, larger seed samples of 100–300 g ensure a more robust and representative analysis. This flexibility supports the analysis of single plant samples up to plot-level samples. Moreover, analysis of mineral content and composition in whole quinoa seeds is under development using energy-dispersive X-ray spectroscopy. Sample organization and tracking is maintained throughout the Nutritional Phenotyping Pipeline using a system that relies on 2D digital barcodes (QR codes), barcode scanners, and USB drivers. The system is designed for quality assurance and quality control by automating sample data entry and processing. Additional information on the pipeline workflow and development as well as videos detailing each step in the process can be found on the SSSL YouTube channel. Available online: https://youtube.com/playlist?list=PLdKoK4IZoGTAFYZWCev4vteOErik3CKcS (accessed on 15 August 2021) 

#### 9.2.4. Detection of Saponins in Quinoa

Most quinoa seeds also contain a large variety of compounds called saponins, some of which have shown to harbor antinutritive properties, thus making them undesirable for human consumption. The predominant form of saponins in quinoa are triterpenoid glycosides [207,208,209]. Saponins have foaming characteristics and are bitter in taste. They have been found to largely localize on the outside of the seed, making it possible to wash them off or remove them by abrasion before consumption (reviewed in [210]). The production of quinoa without those bitter saponins has been a breeding target, and some naturally non-bitter quinoa accessions have been identified [15]. For breeding and phenotyping purposes, testing for saponins is desirable. Over 90 different saponins have been found in quinoa [208]. Saponins are composed of an aglycone backbone with sugar moieties. The combination of those two make up the large variety. Four of the common aglycones in quinoa are oleanolic acid, hederagenin, phytolaccagenic acid, and serjanic acid [208]. 

For phenotyping purposes, a simple detection test may be used that utilizes the foaming characteristics of saponins. When shaken in water, saponins foam, a property that is used for the afrosimetric method [211]. Colorimetric methods such as the use of spectrophotometry can also be used for saponin detection (e.g., [212]). However, these methods also detect the phytosterols in plants. Colorimetric methods are, therefore, not worth the extra effort, while it is easier and cheaper to do the afrosimetric test. Jarvis et al., (2017) [15] validated the results of an afrosimetric test using a more specific detection method, i.e., gas chromatography–mass spectrometry (GC–MS), on a mapping population, which segregated for saponins. The afrosimetric test and GC–MS method corroborated the absence or presence of saponins. 

The afrosimetric test can be easily performed as part of a field trial on a large number of samples according to the following method: five quinoa seeds, free from loose hull, are placed in a 1.5 mL microcentrifuge tube containing 500 µL of double-distilled water. The tubes are shaken by hand vigorously for 30 s until a foam appears that is stable in height, as shown in Figure 10. If foam occurs, it may be semi-quantified by using a caliper and measuring the foam height. A vortex swirls the sample and does not, on its own, lead to stable foam heights. The afrosimetric method is not suitable for the quantification of saponins or investigation of the types of saponins. 

To quantify saponins, GC–MS or liquid chromatography–mass spectrometry (LC–MS) can be used with appropriate standards. GC–MS includes a derivatization step, which removes the sugar moieties from the aglycone backbone; hence, GC–MS allows quantification based on the aglycone backbone [15]. LC–MS allows detection of the individual saponins; however, this quantification is limited by the availability of standards. Both GC–MS and LC–MS are more laborious and expensive than the afrosimetric test.

#### 9.2.5. Quinoa Seed Longevity

Every stage of seed production, from field selection to harvesting and processing to seed storage, is crucial for the quality management of seed [213]. Seed longevity is related to the prediction of seed viability in a storage environment, and depends greatly on its composition, the environment during seed maturation, and harvesting [214,215]. Moisture content, temperature, and oxygen are fundamental factors that control seed longevity [216]. Among these factors, elevated seed moisture is the main culprit for loss of seed quality during storage [217,218].

Both natural and economic resources are wasted owing to inadequate seed storage if seeds of poor quality are sown [219]. Thus, high quality seeds are ensured at the time of planting if the seeds maintain their quality during seed production and at the time of harvesting, processing, and storage. Seeds having higher initial quality have greater longevity than seeds from the same genotype of lower initial quality [219]. Environmental conditions, especially higher temperatures and long photoperiods, during seed development promote dormancy after harvest [220,221]. As quinoa belongs to the Amaranthaceae family, its varieties proceed through different types of seed dormancy or sometimes have no dormancy [222]. The influence of maternal environment on the seed-coat’s characteristics is associated with the level of dormancy in Chenopodium seeds. The Chilean accession showing higher level of dormancy had a significantly thicker episperm for all sowing dates [221]. Some quinoa varieties have no dormancy and in wet environments, seeds may germinate inside the panicle before harvest [222]. This condition can lead to large yield losses and the desiccation intolerance of unorthodox seeds leads to different storage requirements. It is, therefore, important that quinoa varieties are also evaluated for their dormancy type and that preharvest sprouting is recorded when observed. After harvest, the viability of seed should be monitored with Tetrazolium tests on regular basis (see [221]).

Cultivated quinoa has small, flat seeds that are highly hygroscopic in nature and absorb water very quickly, within a day, owing to the porosity of its integument. This moisture gain can be used as an indicator to predict seed longevity [220]. Despite the potential to grow quinoa under adverse environmental conditions, its seed quality deteriorates with inadequate storage conditions, particularly at high temperatures and relative humidity [223,224]. Quinoa seed loses viability extremely quickly compared with conventional cereals such as maize, wheat, and rice [225]. There is a need to explore the physiological and biochemical changes associated with seed longevity under ambient storage conditions. Castellión et al., (2010) [226] found a strong association between quinoa seed aging and the accumulation of Maillard reaction products formed by a reaction between amino acids and reducing sugars, which is responsible for protein aggregation and insolubility. Thus, protein insolubility and water mobility through the multilayers of the seeds are key indicators for the prediction of seed longevity in quinoa germplasm [220]. Both pericarp and seed coat are comprised of two layers of cells. In the pericarp, the inner layer is discontinuous and its cells are tangentially stretched while the large cells of the outer layer are papillose in shape. The seed coat also consists of two cell layers, the exotesta and the endotegmen [227]. In contrast, lipid peroxidation is not a good indicator for seed longevity because of the high oxidative stability of the lipids associated with the high vitamin E content of quinoa seeds [228]. 

The pattern of loss of viability among quinoa accessions depends on post-harvest management, seed provenance, germplasm, and conditions prevailing during seed development, where seed maturation is the most sensitive phase for seed viability [222]. Low seed moisture content and temperature are basic principles for the storage of orthodox seeds such as many quinoa varieties [219]. The best approach is Dry Chain Technology, which is aimed at proper drying (natural or artificial drying to safe moisture limits) of seeds after harvest followed by hermetic packaging to keep it dry until used in the value chain [218]. Hermetic bags are composed of a plastic that resists the exchange of moisture and gases, thus a modified atmosphere can be created by depleting oxygen and enriching carbon dioxide inside the bags [217]. The popular hermetic bags are Super Bags (GrainPro, Washington, D.C., USA) and Purdue Improved Crop Storage (PICS) bags, and are being used in >80 countries to protect grains, legumes, and industrial commodities [229]. A PICS bag is comprised of a double layer of high-density polyethylene (HDPE) liners inside of a woven bag, while a Super Bag consists of a single HDPE layer to control post-harvest losses in cereals [230]. In a recent study conducted at agro-climatic conditions of Pakistan, quality of quinoa seeds stored in hermetic bags at 8% initial seed moisture content is preserved in terms of higher germination and vigor and negligible seed deterioration compared with traditional storage under diverse ambient conditions. Rapid loss of seed viability in traditional porous bags was owing to moisture absorption from the ambient high relative humidity, which resulted in seed deterioration [231].

Quinoa seed of initial seed quality of 80% germination and 8% moisture content can be stored hermetically for six months without loss of viability under ambient conditions (25–40 ℃ and 50–60% RH) while after one-year storage germination declines to 15% [231]. Quinoa seed maintains physiological quality for longer periods (up to 300 days) in impermeable packaging and under low temperature (4 ± 2 ℃) [232]. It is also reported that quinoa seed with 5% moisture content at 5 ℃ can be stored for 8 months with maximum viability [233], and this moisture content is also ideal for long term storage. Seed of 5% moisture content can be stored for one year at 25 ℃ with only one percent loss of germination (according to seed viability equation given by [234]). The moisture content below which quinoa seed longevity is not further improved lies at 4.1% when stored at 65 ℃ [234].

## 10. Conclusions

Advances in quinoa crop improvement can be accelerated through the development of an international network of quinoa researchers and trial datasets. Collaborations across members of the Global Collaborative Network on Quinoa (GCN-Quinoa). Available online: gcn-quinoa.org (accessed on 15.08.2021) would be highly facilitated if all of members will be able to use the same language, tools and methods for establishing trials, collecting data, and then sharing datasets. This network initiated in 2015 has connected all the participants of different FAO–TCP programs. Now, the GCN-Quinoa links 296 members from all around the world in more than 75 countries. The sharing of data among global and regional research groups allows deeper exploration of each dataset in the context of its environment. Owing to the significant effects of environment and management practices on quinoa phenotypes, the reusability of each dataset greatly depends on the quality of the metadata recorded, including a detailed profile of environmental parameters, and on maintaining as many factors constant as is possible. The most important variables that need consistency among trials are the methods of data collection, which can be achieved by an international agreement on phenotyping methods describing common protocols for establishing a dataset with comparable standards, as have been proposed in this article. The Germinate database is available at Quinoa Germinate Database. Available online: http://germinate.quinoadb.org (accessed on 15 August 2021) to further facilitate standardizing global quinoa dataset structures and sharing and analyzing of data. 

Quinoa is a remarkable crop with many valuable properties, but it is also a crop that still needs significant amounts of research and breeding to facilitate its move to become a major or widely cultivated food crop. It is hoped that this paper will facilitate these efforts by providing a framework for globally consistent phenotyping, benchmarking the phenotyping of quinoa plants and easing the comparison of results obtained around the world.

## Figures and Tables

**Figure 1 plants-10-01759-f001:**
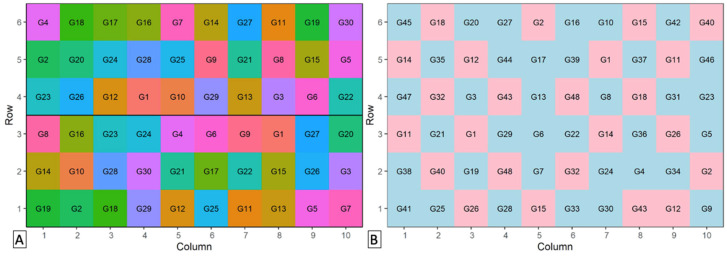
Example trial design generated using DiGGer of (**A**) a fully replicated trial with two replicates, including 60 plots arranged in six rows by 10 columns, with each replicate block of 30 genotypes (GXX) comprising three rows by 10 columns (color representing genotype), and (**B**) a partially replicated trial with 25% replication, including 48 genotypes (GXX) in total, with 12 genotypes having two replicates (pink) and 36 genotypes having one replicate (blue).

**Figure 2 plants-10-01759-f002:**
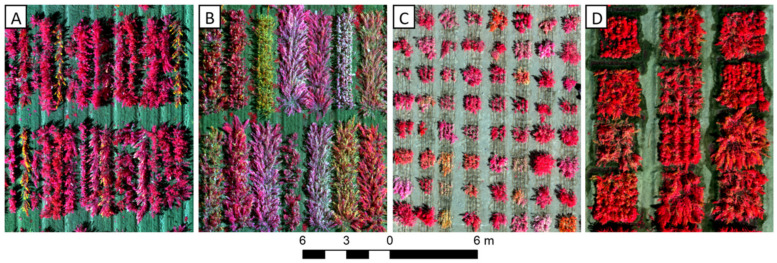
Quinoa plant trials showing (**A**) linear plots with a width of one plant; (**B**) rectangular plots with multiple plants next to each other; (**C**) square plots of 1 × 1 m^2^; and (**D**) square plots of 2.5 × 2.5 m^2^. All UAV images are displayed as false color composites.

**Figure 3 plants-10-01759-f003:**
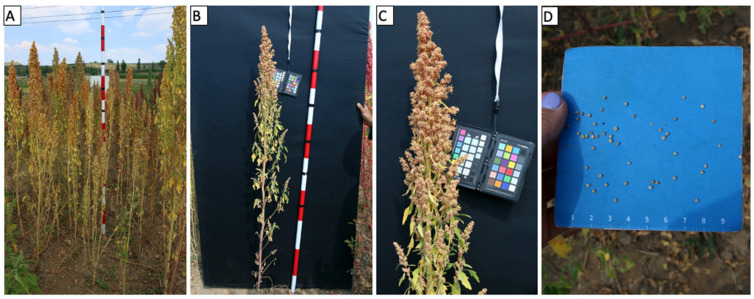
Example photos of images taken during mature plant phenotyping event. These include (**A**) a picture of the entire plot with reference for plant height, (**B**) a photograph of one representative plant for the plot in front of a black background with ColorChecker and ranging pole for height reference, (**C**) a close-up picture of the primary panicle, and (**D**) a picture of a seed sample (~20 seeds) on a 10 × 10 cm square blue card background (used for later image extractions of seed size and color.

**Figure 4 plants-10-01759-f004:**
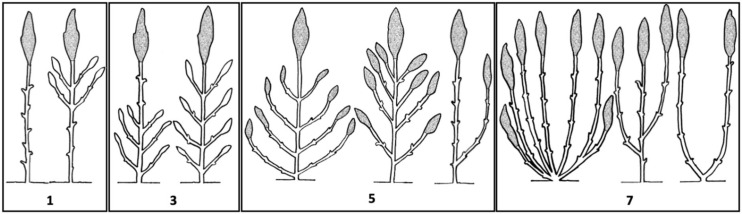
Growth habit of quinoa plants is grouped into four categories based on the branching habits at the base of the plant, and the size of the panicles of these branches: **1**—not branched at base; **3**—some branching from the base with no significant panicles on branches; **5**—branching from the base with more significant panicles; **7**—main panicle is difficult to identify.

**Figure 5 plants-10-01759-f005:**
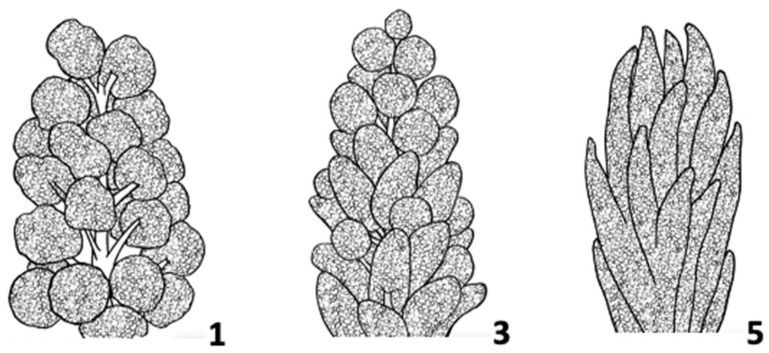
Three distinct groups of observable panicle shapes: **1**. Glomerulate, **3**. Intermediate, and **5**. Amarantiform.

**Figure 6 plants-10-01759-f006:**
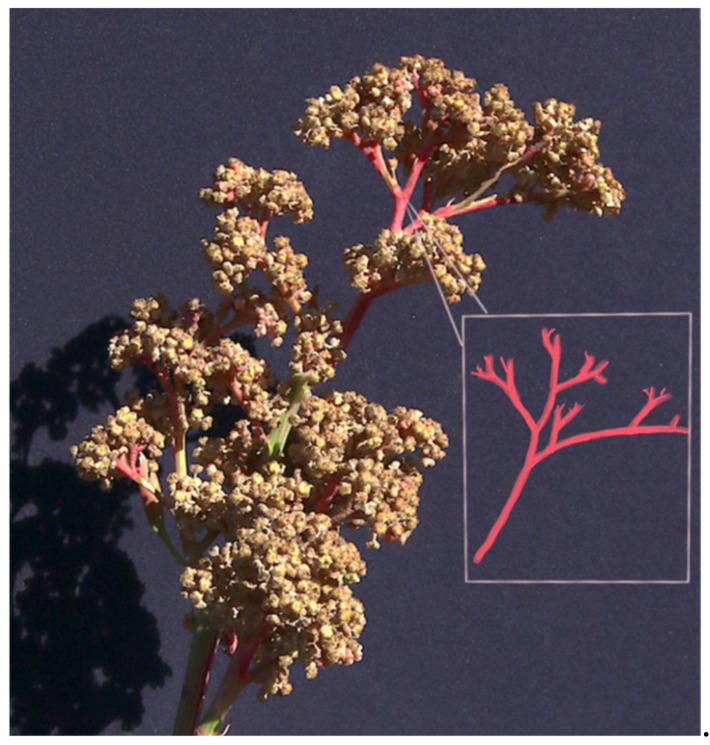
An example of a glomerulate panicle with a visible network of branches.

**Figure 7 plants-10-01759-f007:**
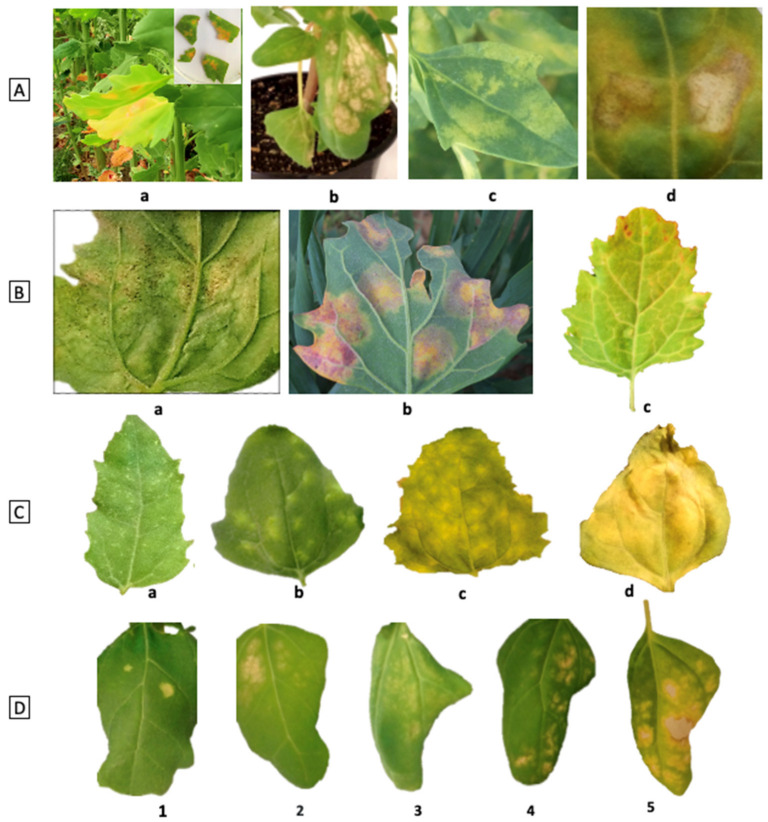
(**A**) Lesions on leaf surface: (**a**) pale or yellow chlorotic lesions with or without a halo and occasional pink-orange discoloration caused by leaf pathogens, (**b**) bronze irregular lesions caused by *Alternaria *sp., (**c**) diffuse chlorotic spots caused by *P. variabilis* and (**d**) concentric and chlorotic halo under artificial inoculation with *Alternaria *sp. (**B**) Sporulation on lower side leaf surface with (**a**) black dots showing downy mildew sporangia, (**b**) dark gray-violaceous sporulation caused by *P. variabilis* and (**c**) vein discoloration, general chlorosis and pink-orange spots caused by *Fusarium *sp. (**C**) Lesion type on upper surfaces and amount of disease ranging from (**a**) dots, (**b**) dots expanding, (**c**) diffuse, and (**d**) extensive. (**D**) Severity phenotyping scale used for assessing the percentage of the leaf area affected where 1 = 0–10%, 2 = 11–25%, 3 = 26–50%, 4 = 51–70% and 5 = 71–100%. Leaf examples given represent different degrees of severity during the infection of *Alternaria *spp. (Colque-Little and some images previously published in [51]).

**Figure 8 plants-10-01759-f008:**
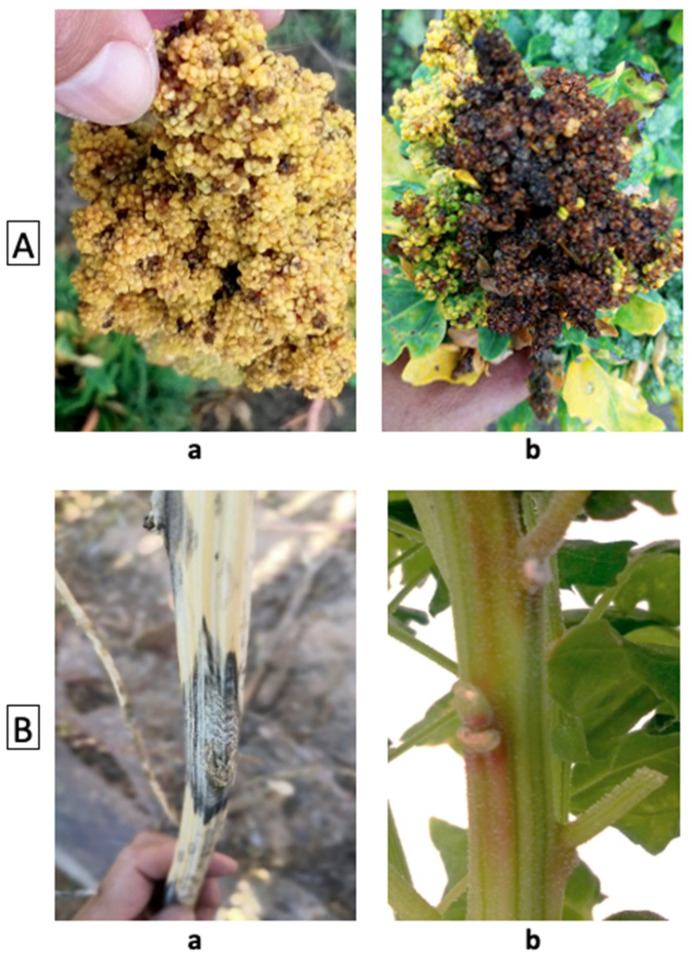
(**A**) Examples for panicle diseases, with (**a**) a panicle infected with *Alternaria spp*., and (**b**), a panicle predominantly infected with *Cladosporium *spp. at the end of the season. (**B**) Examples for stem diseases, showing (**a**) quinoa black stem caused by *Ascochyta caulina* with presence of dark structures (pycnidia), (**b**) pink stem and light pink mycelia corresponding to *Fusarium* spp. (pictures by Colque-Little, (**B**)(a) courtesy of [182].

**Figure 9 plants-10-01759-f009:**
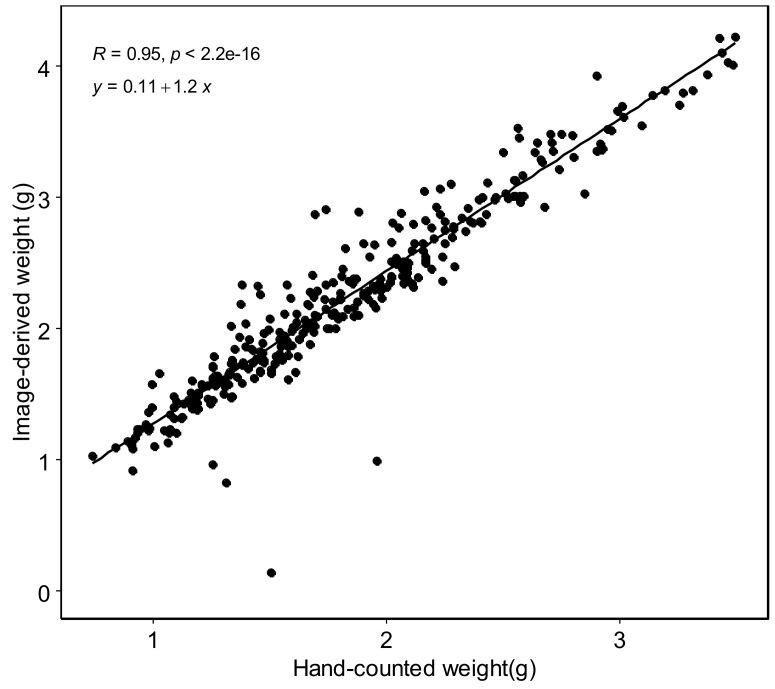
Correlation between thousand seed weight as estimated using image analysis, where the number of seeds is counted and the weight of those seeds is known, and hand-counted thousand seed weight.

**Figure 10 plants-10-01759-f010:**
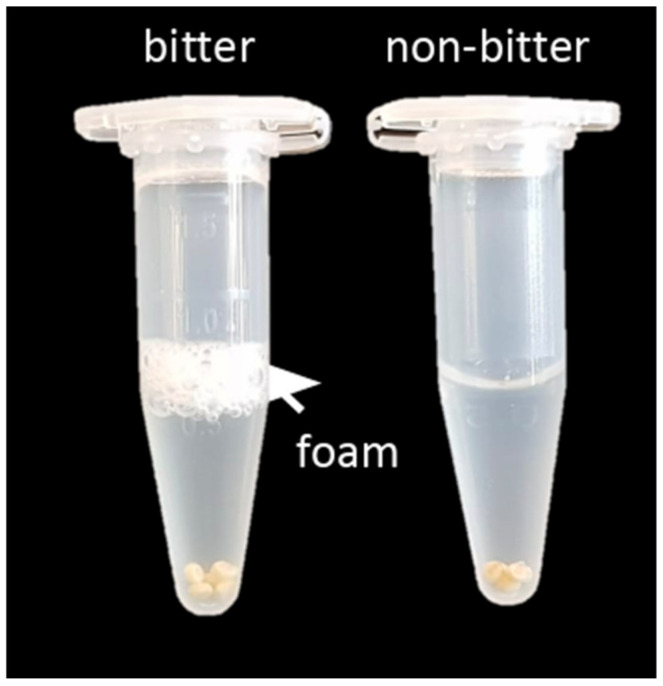
Foaming test (afrosimetric method) for presence or absence of saponins on quinoa seeds.

**Table 1 plants-10-01759-t001:** Minimum set of environmental variables to record for quinoa trials.

Soil, To Be Measured Before and After the Field Season
Watering regime
Water holding capacity
Composition in terms of % sand, silt, organic matter, etc.
Nutrient and mineral composition—total nitrogen, organic carbon, phosphorus, potassium, sulfur, etc. Note: when measuring nitrates, the soil sample must be kept cold because nitrates are unstable
Soil physical properties affecting plant growth
pH
Apparent density
Electrical conductivity (EC), especially for salinity trials
**Weather**
Precipitation, and irrigation schedule
Temperature, at least daily T_max_ and T_min_, but preferably recorded continuously throughout the day to enable calculation of degree-days to flowering and to maturity
Humidity—relative humidity/dewpoint temperature
Daily irradiance (mol m^−2^ d^−1^), recorded continuously throughout the day
Wind speed (average daily speed)
Day length (including twilight time)

**Table 2 plants-10-01759-t002:** Overview and description of phenotypes that may be recorded at the plot level and record the majority of the plot. Scoring metrics followed by an asterisk(*) represent assessments of the percentage of the plot covered or affected by levels 1 (up to 20%), 3 (20–40%), 5 (40–60%), 7 (70%–80%), and 9 (over 80%). Border plants of each plot should not be considered when phenotyping. See Appendix A cards for visual examples for the traits.

Plot-Level Phenotype	Scoring Metric	Description
Plot coverage	1,3,5,7,9 *	Percentage of the plot covered, 1 = poor to 9 = good establishment
Plot population homogeneity	1,3,5,7	Judgment of homogeneity of the accession, 1 = homogeneous to 7 = mixed
Branchiness	1,3,5,7	Score for the overall amount of side branches along the entire length of the stem, ignoring very small and spindly branches, ranging from 1 = no branches to 7 = bushy plant with many (i.e., greater than 7) major lateral branches
Growth habit	1,3,5,7	Four categories of growth habit described in images on the phenotyping card. Here the focus lies on whether branching is present in the bottom third of the stem from the base of the plant and if a main inflorescence can be identified
Stem breakage incidence	1,3,5,7,9 *	Stems are broken or detached, assessing the percentage of the plot affected
Stem lodging incidence	1,3,5,7,9 *	Plants are prostrate, on or near the ground, with intact stems; assessing the percentage of the plot affected
Stem lying incidence	1,3,5,7,9 *	Stem of the plant is not emerging straight up from the soil but has a kink at the base, growing along the ground before rising; assessing the percentage of the plot affected
Stem angle	1,3,5,7	The angle at which the majority of plants are leaning, measured between the vertical axis and the horizontal axis
Panicle axis angle	1,3,5,7	The angle at which the majority of panicle axes are leaning, measured between an upright panicle on the vertical and a panicle pointing towards the ground

**Table 3 plants-10-01759-t003:** Overview of the quantitative plant-level phenotypes that are measured for representative plants of a plot.

Plant-Level Phenotype	Unit	Description
Plant height	cm	Height of the most representative plants of the plot, usually from the middle of the plot, measured with a long measuring stick from soil to the tip of the panicle. If more than one distinct phenotype is present, more than one plant may be recorded in a new row of the spreadsheet, with all phenotypes that are differing recorded separately
Panicle length	cm	Length of the primary panicle measured with the same stick. Measured from the base of the panicle to the tip
Stem diameter near plant base	mm	Thickness of the stem measured with calipers at the middle of the bottom third of the plant stem
Stem diameter under panicle	mm	Thickness of the stem measured just underneath the panicle
Number of significant panicles	count	Count of the number of significant panicles, i.e., larger panicles, near the top of the plant, harvestable, that provide a major contribution to the seed harvested from the plant

**Table 4 plants-10-01759-t004:** Overview of categorical plant-level phenotypes and description of the according scoring metrics.

Plant-Level Phenotype	Scoring Metric	Description
Growth stage	BBCH scale	Phenological growth stage; very important to record at mature phenotyping
Seed shattering	1,3,5,7	Grain persistence in the plant at physiological maturity. Assessing how easy seeds fall off the panicle upon light touch: 1 = no seeds falling to 7 = majority of seeds falling
Panicle shape	1,3,5	Classified into one of the three categories: glomerulate, intermediate, or amarantiform
Panicle density	1,3,5,7	Scored from 1 = lax (loose) with panicle axes easily visible to 7 = tight and compact panicles
Panicle leafiness	1,3,5,7	Scored from 1 = no leaves to 7 = many leaves
Panicle color	13,4,15,16,5,7	Categorized according to the color phenotyping card
Stem color	13,4,15	Categorized into green (13), red (4), or no pigmentation (15)
Stem striae	0,1	Presence (1) absence (0) scoring of stem streaks or stripes
Axil pigmentation	0,1	Presence (1) absence (0) scoring of pigmented axils
Stem leaf shape	1,2	Leaves of the stem are categorized into two groups: rhomboidal (1) and triangular (2)

**Table 5 plants-10-01759-t005:** Overview of harvest and post-harvest traits.

Harvest and Post-Harvest	Unit	Description
Number of plants harvested	count	
Above-ground dry biomass	grams	Cutting plants at the very base with secateurs and drying the entire plant in an oven until mass is constant. Recording total dry weight
Below-ground biomass	grams	If possible, root biomass could also be measured (especially when plants are growing in sandy soil)
Seed yield for representative plants	grams	Seed mass of approximately four representative plants that were harvested from the center of the plot (seed should be dried to constant weight)
Total seed yield per plot	grams × m^−2^	Harvesting the panicles remaining per plot while excluding borders, and adding the weight to that from the four representative plants, seed dried in oven to constant weight
Seed yield per plant	grams	Total harvested seed mass per plant may be calculated from the seed weight of all plants in the plot divided by the number of plants harvested
Harvest index		Yield/ above-ground biomass
Seed weight (TGW)	grams/1000 seeds	Thousand Grain Weight (TGW), the weight of 1000 seeds
Seed hectoliter weight	grams/100mL	Estimation of density, determined by weighing all seeds fitting into a 100 mL volume
Seed size (average area; average perimeter)	millimeter	Seed size outputs from image analysis separated by semicolon (method options described in Section 9.2)
Seed color (average red; average green; average blue)	Numeric RGB equivalent	Seed color output values for red, green, and blue components, semi colon separated (obtained from image analysis methods, see Section 9.2)

**Table 6 plants-10-01759-t006:** Washington State University Sustainable Seed Systems lab NIR calibration (V3) metrics. The range, minimum (min), and maximum (max) are calculated using reference data for quinoa samples included in the calibration (*n* = 175). Calibration prediction accuracy metrics are reported as an average measure of 8-fold cross validation in triplicate.

	Stats from WSU Calibration V3 Data (g 100g^−1^ Protein)							
	Range	Min	Max	RMSECV	SECV	Robust SECV	RPDCV	R2CV
Alanine	1.99	2.89	4.88	0.022	0.022	0.018	3.036	0.892
Arginine	4.68	4.58	9.25	0.053	0.053	0.044	4.308	0.946
Aspartic acid	3.22	5.51	8.73	0.039	0.040	0.036	3.768	0.930
Cysteine	0.76	1.31	2.07	0.010	0.010	0.010	3.188	0.902
Glutamic acid	7.04	8.22	15.26	0.093	0.093	0.086	3.802	0.931
Glycine	1.33	4.78	6.11	0.041	0.041	0.036	2.447	0.834
Histidine	1.05	1.96	3.01	0.015	0.015	0.014	4.564	0.952
Isoleucine	1.51	2.89	4.41	0.021	0.022	0.019	3.392	0.913
Leucine	2.55	4.3	6.85	0.031	0.032	0.029	3.473	0.917
Lysine	3.14	3.45	6.59	0.029	0.029	0.033	3.290	0.908
Methionine	1.15	1.31	2.46	0.012	0.012	0.009	2.955	0.886
Phenylalanine	1.57	2.71	4.28	0.019	0.019	0.018	3.889	0.934
Proline	1.68	2.80	4.48	0.023	0.023	0.018	2.556	0.847
Serine	1.39	2.89	4.28	0.019	0.019	0.016	3.176	0.901
Taurine	1.96	0.82	2.79	0.012	0.012	0.009	1.669	0.645
Threonine	1.60	2.43	4.02	0.017	0.017	0.016	3.015	0.890
Tryptophan	0.93	0.55	1.48	0.012	0.012	0.009	1.681	0.647
Tyrosine	0.93	2.12	3.05	0.014	0.014	0.013	3.393	0.913
Valine	1.84	3.36	5.20	0.024	0.024	0.023	3.260	0.906
Hydroxylysine	0.18	0.05	0.23	0.004	0.004	0.003	1.591	0.605
Hydroxyproline	0.93	0.29	1.21	0.010	0.010	0.011	1.821	0.699
**Stats from WSU calibration V3 data (g 100g^−1^ sample)**								
Crude protein	11.95	6.82	18.77	0.394	0.395	0.406	5.521	0.967
Ash	3.32	2.21	5.53	0.154	0.154	0.129	3.084	0.895
Crude fat	6.95	0.00	6.95	0.310	0.311	0.316	3.883	0.934
Crude fiber	13.67	1.44	15.11	0.442	0.443	0.377	4.904	0.958
Moisture	3.76	6.41	10.17	0.183	0.183	0.159	6.579	0.977
TotalAA	10.06	5.84	15.90	0.413	0.413	0.328	4.018	0.938
	Range	Min	Max	RMSECV	SECV	Robust SECV	RPDCV	R2CV

Hydroxylysine and hydroxyproline are poorly predicted. Lanthionine and ornithine were eliminated from the calibration owing to limited lab analysis. RMSECV = root mean square error of cross validation; SECV = standard error of cross validation; RPDCV = ratio of reference data standard deviation to standard error of prediction; R2CV = coefficient of determination of cross validation.

## Data Availability

Data available upon request.

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
