# Peer review of "Quinoa Phenotyping Methodologies: An International Consensus"

_plants, 2021, doi:10.3390/plants10091759_

Round 1

Reviewer 1 Report

The review by Stanschewski et al. is an extensive guide to "phenotyping" methodologies that can or should be used in quinoa field trials. The manuscript provides extensive information and recommendations regarding every aspect of setting up and performing field trials in this crop, as well as analyzing yield and measuring important traits post-harvest. Also, the review addresses appropriately the necessity to adopt a common language in defining phenotypic traits, metadata, etc by referring to existing ontologies and data standards. These aspects ensure that data re-use is made available to the largest possible community also outside quinoa researchers, strictly speaking. It is likely that this review could be read with interest by a number of quinoa researchers, especially young researchers. 

There are a few aspects that could be considered to further improve the manuscript:

  • the writing is somewhat verbose. The total length could be reduced (e.g., 15-20%) without loss of messages.
  • many sections of the review are fully or in part generic and they are applicable to many crops, for example, on experimental design, or remote sensing phenotyping using UAVs. Any effort to shorten these sections or make them more specific to quinoa could improve the manuscript.
  • the Germinate website apparently does not contain accessible data yet, or does it require registration before one can access the datasets. In any case, it would be of great utility to provide exemplary data on the website accompanying this review.
  • the terminology used to foster comparability of results of different filed trials can be misleading. Here are some examples, which the authors may debate further. 1) page 4, lines 182-186. Speaking of "error (noise)" does not seem appropriate. The opinion of this reviewer is that if there are no good guidelines for recording traits or there is margin for interpretation, there is uncertainty which may lead to exclude some data sets altogether. However, it does not mean that there is necessarily an error or (biological? technical?) noise; 2) page 5, lines 216-218. Speaking of standardization might be pushing it too far concerning field trials. Perhaps it would be better to say that all the field trials should be conducted according to the guidelines, or just using standardized templates, as stated later on in the text. 3) page 13. line 648. "Several environmental variables cannot be controlled in a field experiment". In reality, this should be read as "virtually none". 4) page 14, line 660. "normalized" is misleading referred to management practices. Harmonized by adopting the guidelines seems more realistic and appropriate.  

Author Response

Dear reviewer,

Thank you very much for your nice feedback and for your valuable time that you invested for taking such a detailed look at the manuscript. Please find our responses to your comments in purple text colour.

Kind regards,

Clara and Mark

Reviewer 1.

The review by Stanschewski et al. is an extensive guide to "phenotyping" methodologies that can or should be used in quinoa field trials. The manuscript provides extensive information and recommendations regarding every aspect of setting up and performing field trials in this crop, as well as analyzing yield and measuring important traits post-harvest. Also, the review addresses appropriately the necessity to adopt a common language in defining phenotypic traits, metadata, etc by referring to existing ontologies and data standards. These aspects ensure that data re-use is made available to the largest possible community also outside quinoa researchers, strictly speaking. It is likely that this review could be read with interest by a number of quinoa researchers, especially young researchers. 

There are a few aspects that could be considered to further improve the manuscript:

  • the writing is somewhat verbose. The total length could be reduced (e.g., 15-20%) without loss of messages.

Response: We agree that the manuscript is of unusual length and could be reduced. In that process we would however remove a lot of the detail in our descriptions that we included on purpose. We find that this level of detail supports the comparability in how experiments and data collection is conducted, especially because the quinoa community includes a lot of people with English as a second language.

  • many sections of the review are fully or in part generic and they are applicable to many crops, for example, on experimental design, or remote sensing phenotyping using UAVs. Any effort to shorten these sections or make them more specific to quinoa could improve the manuscript.

Response: The experimental design section can be applicable to many crops and we have tried to include all information that we thought necessary for early career researchers to get started and to yield comparable results. We could not find a way to make it more quinoa-specific.  Similarly for the UAV section, going into too much detail for providing practical information and guidance for UAV data collection and processing for mapping quinoa will be a paper in itself. While the UAV section is somewhat general, it provides references to suitable papers for more details. We hope that this can be acceptable for the journal.

  • the Germinate website apparently does not contain accessible data yet, or does it require registration before one can access the datasets. In any case, it would be of great utility to provide exemplary data on the website accompanying this review.

Response: Agreed, we are making every effort to get the database populated, and have made a first dataset accessible to the public.

  • the terminology used to foster comparability of results of different filed trials can be misleading. Here are some examples, which the authors may debate further.
  • 1) page 4, lines 182-186. Speaking of "error (noise)" does not seem appropriate. The opinion of this reviewer is that if there are no good guidelines for recording traits or there is margin for interpretation, there is uncertainty which may lead to exclude some data sets altogether. However, it does not mean that there is necessarily an error or (biological? technical?) noise;

Response: Thank you, that is a good point. We have made the changes.

  • 2) page 5, lines 216-218. Speaking of standardization might be pushing it too far concerning field trials. Perhaps it would be better to say that all the field trials should be conducted according to the guidelines, or just using standardized templates, as stated later on in the text.

Response: We agree and have adapted the statement in the text: “As quinoa research is progressing, these scales will need to be adapted to match new situations and applications. For this, we are aiming for the Quinoa Phenotyping Consortium to hold an annual meeting to refine guidelines and procedures with the aim of both increasing the quality and standardization of phenotyping.”

  • 3) page 13. line 648. "Several environmental variables cannot be controlled in a field experiment". In reality, this should be read as "virtually none".

Response: Thank you, we made the change.

  • 4) page 14, line 660. "normalized" is misleading referred to management practices. Harmonized by adopting the guidelines seems more realistic and appropriate.

Response: Thank you, we made the change.

Reviewer 2 Report

Comments:

Complete bibliographic review, well written and pleasant to read. The idea of ​​creating a data platform that encourages interaction between researchers is excellent. It describes high throughput methodologies including post-harvest phenotyping and associated statistical analysis. Discusses gemoplasm and plant management from planting to post harvest. It includes nutritional aspects and even saponin. It becomes a reference document for quinoa and other pseudo cereals.

I include some suggestions described below.

Line 76. High troughput seed phenotyping

Line 81. Food systems are experiencing intense pressure owing to, among other factors, increasing populations, migration of food crops to marginal areas due to species for energy production and environmental change…

L363. ...at multiple locations. The use of double haploids to obtain homozygosity as it is done in corn and wheat could also be an option.

L498. Specific response to stress as tested by Jayme-Oliveira et all 2017 and El-Moghazi 2017.

Adilson Jayme-Oliveira, Walter Quadros Ribeiro Júnior, Maria Lucrécia Gerosa Ramos, Adley Camargo Ziviani and Adriano Jakelaitis. Amaranth, quinoa, and millet growth and development under different water regimes in the Brazilian Cerrado.  Pesq. agropec. bras., Brasília, v.52, n.8, p.561-571, ago. 2017 DOI: 10.1590/S0100-204X2017000800001

  1. M. M. Al-Naggar, R. M. Abd El-Salam, A. E. E. Badran and Mai M. A. El-Moghazi. Drought tolerance of Five Quinoa (Chenopodium quinoa Willd.) Genotypes and Its Association with Other Traits under Moderate and Severe Drought Stress., AJAAR, 3(3): 1-13, 2017; Article no. AJAAR.37216.

L520. to be recorded. This is common in quinoa that has recalcitrant seeds.

L575. (Garcia et al, 2003 and Jayme-Oliveira 2017)

L714. …traits are measured. Sensors measuring soil humidity frequently are overestimating soil humidity in soils with high level or iron (Kargas et al 2020) and in this case gravimetric humidity can be done collecting soil samples.

Kargas, G., LONDRA, O., Anastasatou, M., Moustakas,N.  The Effect of Soil Iron on the Estimation of Soil Water Content Using Dielectric Sensors. Water 2020, 12(2), 598; https://doi.org/10.3390/w12020598.

Jackson et al. (1981) created the crop water stress index, which normalizes leaf temperature using also environmental conditions around the experiments, using leaf temperature to wet and dry reference surfaces.

Jackson, R. D., S. B. Idso, R. J. Reginato, and P. J. Pinter. 1981. Canopy temperature

as a crop water stress indicator. Water Resources Research 17(4): 1133–1138.

doi:10.1029/WR017i004p01133.

Line 101. (Galli et al 2020).  

Line 105. large phenotyping event. In this phenological stage, a pigment extraction like chlorophyll could summarize the photosynthetic process that occurred in the previous phases.

L108. …persist. In the need to use several people for evaluations, each one must evaluate an entire repetition and thus identify the interpretive differences in the statistical analysis as a block effect.

Line 114. …2002). In soybeans for example, at 0.9 m away in the Mississippi Delta was 0.41% .

Jeffery D RayThomas C KilenCraig AbelCraig AbelRobert L. ParisRobert L. Paris. Soybean natural cross-pollination rates under field conditions.  Environmental Biosafety Research 2(2):133-8, 2003.

Line 124:  The use of growth regulators as well as trinexapac ethil may be an option in the case of a region with a lodging historic although experiments with doses and time of application are yet to be conducted.ne 124. T

Additional comments:

1) It is not uncommon for rains to occur in the harvest that cause the germination of grains in the panicles. A sub-item could be added to that effect. See the picture.

2) Quinoa has been tested as a cover plant compared to Amaranth and millet (Jayme-Oliveira et al 2017). Something in that sense could be put in the text.

Adilson Jayme-Oliveira, Walter Quadros Ribeiro Júnior, Maria Lucrécia Gerosa Ramos, Adley Camargo Ziviani and Adriano Jakelaitis. Amaranth, quinoa, and millet growth and development under different water regimes in the Brazilian Cerrado. Pesq. agropec. bras., Brasília, v.52, n.8, p.561-571, ago. 2017.

Author Response

Dear reviewer,

Thank you very much for your nice feedback and for your valuable time that you invested for taking such a detailed look at the manuscript. Please find our responses to your comments in purple text colour.

Kind regards,

Clara and Mark

Reviewer 2.

Complete bibliographic review, well written and pleasant to read. The idea of ​​creating a data platform that encourages interaction between researchers is excellent. It describes high throughput methodologies including post-harvest phenotyping and associated statistical analysis. Discusses gemoplasm and plant management from planting to post harvest. It includes nutritional aspects and even saponin. It becomes a reference document for quinoa and other pseudo cereals.

I include some suggestions described below.

  • Line 76. High troughputseed phenotyping

Response: Thank you, we have added high throughput seed phenotyping to the keywords of the manuscript. 

  • Line 81. Food systems are experiencing intense pressure owing to, among other factors, increasing populations, migration of food crops to marginal areas due to species for energy productionand environmental change…

Response: Thank you for your suggestion. This is the first sentence of the introduction, therefore we would like to keep the focus on the major factors affecting our food systems, such as a growing world population and climate change. With “, among other factors,” we did acknowledge that there are other factors adding to this. 

  • ...at multiple locations. The use of double haploids to obtain homozygosity as it is done in corn and wheat could also be an option.

Response: Please see line 358-360, we had already mentioned this in the text (at a different location than suggested).

  • Specific response to stress as tested by Jayme-Oliveira et all 2017 and El-Moghazi 2017.

Adilson Jayme-Oliveira, Walter Quadros Ribeiro Júnior, Maria Lucrécia Gerosa Ramos, Adley Camargo Ziviani and Adriano Jakelaitis. Amaranth, quinoa, and millet growth and development under different water regimes in the Brazilian Cerrado.  Pesq. agropec. bras., Brasília, v.52, n.8, p.561-571, ago. 2017 DOI: 10.1590/S0100-204X2017000800001

  1. M. Al-Naggar, R. M. Abd El-Salam, A. E. E. Badran and Mai M. A. El-Moghazi. Drought tolerance of Five Quinoa (Chenopodium quinoa Willd.) Genotypes and Its Association with Other Traits under Moderate and Severe Drought Stress., AJAAR, 3(3): 1-13, 2017; Article no. AJAAR.37216.

Response: “If stress effects are important, these conditions must be implemented in plants grown under otherwise optimal conditions to allow quantification of the specific responses to stress.”
This statement is a general statement that needs to be followed in any trial assessing plant stress. The suggested citations are examples for trials that tested certain stress responses in quinoa, but are not specifically highlighting a general method to follow. This section is about trial design, and not a review about quinoa stress response.

  • to be recorded. This is common in quinoa that has recalcitrant seeds.

Response: The sentence of the manuscript at which the insertion was suggested is the following: “Considering the significant phenotypic plasticity of quinoa in response to planting density, it is clear that trials with similar planting densities are better in multi-environment analyses and that planting density always needs to be recorded.”
We do not believe that quinoa varieties with recalcitrant seeds need to be highlighted in this context. These varieties, when grown in wet environments, may have the issue of preharvest sprouting, but the seedlings are not plants that would be counted when assessing planting density (as these seedlings are not affecting mature plant phenotypes of the mature plants in the plot).

  • (Garcia et al, 2003 and Jayme-Oliveira 2017)

Response: The sentence of the manuscript at which the insertion was suggested is the following: “Irrigation is known to affect several aspects of quinoa phenotypes, from plant height (Yang et al., 2016) to seed saponin content (Präger et al., 2018) and yield (Garcia et al., 2003).”

We agree that the citation from Jayme-Oliveira et al. 2017 that was listed above can be cited here, because grain weight was a trait measured in a trial comparing different water regimes, and we have included it.

  • …traits are measured. Sensors measuring soil humidity frequently are overestimating soil humidity in soils with high level or iron (Kargas et al 2020) and in this case gravimetric humidity can be done collecting soil samples.

Kargas, G., LONDRA, O., Anastasatou, M., Moustakas,N.  The Effect of Soil Iron on the Estimation of Soil Water Content Using Dielectric Sensors. Water 2020, 12(2), 598; https://doi.org/10.3390/w12020598.

Response: Please see line 724-728, this had already been addressed at a different position than suggested.

  • Jackson et al. (1981) created the crop water stress index, which normalizes leaf temperature using also environmental conditions around the experiments, using leaf temperature to wet and dry reference surfaces.

Jackson, R. D., S. B. Idso, R. J. Reginato, and P. J. Pinter. 1981. Canopy temperature as a crop water stress indicator. Water Resources Research 17(4): 1133–1138. doi:10.1029/WR017i004p01133.

Response: Please see line 951-953, we are already referring to Jackson et al 1981.

  • Line 101. (Galli et al 2020).  

Response: I believe there is mistake in the line references from this point on. Could it be that you are referring to line 1014? Here, Galli et al 2020 is already referenced, I don’t understand this comment [“While in situ calibration data allow relationships and models to be developed, e.g. for UAV-based image classification, independent validation data enable the accuracy of maps to be assessed (Galli et al., 2020)”]

  • Line 105. large phenotyping event. In this phenological stage, a pigment extraction like chlorophyll could summarize the photosynthetic process that occurred in the previous phases.

Response: Line 1055. While we do say: “Phenotyping at this stage can be seen as the cumulative effect of different developmental phases on the trait studied.”, we believe that a trait such as chlorophyll concentration is not one of those traits and chlorophyll concentration at maturity is not necessarily summarizing photosynthetic processes of a plant during development. This is because rates of photosynthesis can change quickly in response to environmental conditions. For this reason, chlorophyll fluorescence is one of traits used to identify early plant stress (https://doi.org/10.15835/nbha48412059 ) - and chlorophyll fluorescence can correlate with chlorophyll concentration (https://link.springer.com/article/10.1007/s11099-005-0161-4).

  • …persist. In the need to use several people for evaluations, each one must evaluate an entire repetition and thus identify the interpretive differences in the statistical analysis as a block effect.

Response: We agree that care is required and we elaborated a strategy in this paragraph (1072-1083). Including multiple people for evaluations is a necessity that arises from time restrictions in large trials. From our experience, it is not feasible to let each person perform an entire repetition and performing statistical analyses to identify differences in scoring in a field situation. We believe that it is sufficient to score overlapping plots and comparing results until all people are in agreement, and to repeat these checkpoints multiple times throughout each day. 

  • Line 114. …2002). In soybeans for example, at 0.9 m away in the Mississippi Delta was 0.41% .

Jeffery D RayThomas C KilenCraig AbelCraig AbelRobert L. ParisRobert L. Paris. Soybean natural cross-pollination rates under field conditions.  Environmental Biosafety Research 2(2):133-8, 2003.

Response: Line 1143. Due to great differences between different species, we do not find it necessary to refer to natural cross-pollination rates of soybeans in the context of natural hybridization rates of quinoa. 

  • Line 124:  The use of growth regulators as well as trinexapac ethil may be an option in the case of a region with a lodging historic although experiments with doses and time of application are yet to be conducted.

Response: We do not see how this comment is related to a paper focusing on phenotyping methodologies.

Additional comments:

  • It is not uncommon for rains to occur in the harvest that cause the germination of grains in the panicles. A sub-item could be added to that effect. See the picture.

Response: Thank you for pointing this out, we were not aware of the importance of this trait in some regions. We have added a few sentences in the text (Line 1877-1884).

“Quinoa seeds vary in their characteristics depending on variety, with some classified as orthodox seeds that have dormancy and be dried to 5% moisture, while other quinoa varieties have unorthodox seeds. Unorthodox seeds have no dormancy and in wet environments, seeds may germinate inside the panicle before harvest (McGinty et al., 2021). This condition can lead to large yield losses and the desiccation intolerance of unorthodox seeds leads to different storage requirements. It is therefore important that quinoa varieties are also evaluated for their dormancy type and that preharvest sprouting is recorded when observed.”

  • Quinoa has been tested as a cover plant compared to Amaranth and millet (Jayme-Oliveira et al 2017). Something in that sense could be put in the text.

Adilson Jayme-Oliveira, Walter Quadros Ribeiro Júnior, Maria Lucrécia Gerosa Ramos, Adley Camargo Ziviani and Adriano Jakelaitis. Amaranth, quinoa, and millet growth and development under different water regimes in the Brazilian Cerrado. Pesq. agropec. bras., Brasília, v.52, n.8, p.561-571, ago. 2017.

Response: We do not agree with the suggestion of discussing the use of quinoa as a cover plant in this manuscript, as the aim of this paper is not the review of quinoa uses and applications.

Round 2

Reviewer 1 Report

The authors have addressed the points that were raised. In particular, it is good that exemplary data are now contained in the Germinate database. Although the manuscript is still considerably long and overall generic, i.e., it does not only apply to quinoa experimental work but to many other crops, too, the authors explained that it is their intention to provide guidelines as detailed as possible.